Detailed anatomy of the braincase of Macelognathus vagans Marsh, 1884 (Archosauria, Crocodylomorpha) using high resolution tomography and new insights on basal crocodylomorph phylogeny

Leardi Juan Martin jmleardi@gl.fcen.uba.ar 1
Pol Diego 2
Clark James Matthew 3
1 CONICET, Instituto de Estudios Andinos “Don Pablo Groeber” (IDEAN), Facultad de Ciencias Exactas y Naturales, Departamento de Ciencias Geológicas, Universidad de Buenos Aires , Buenos Aires , Argentina
2 CONICET, Museo Paleontológico Egidio Feruglio , Trelew , Chubut , Argentina
3 Department of Biological Sciences, George Washington University , Washington, D.C. , United States of America
Young Mark
Electronic publication date: 2017 Jan 19
Publication date: 2017
Volume: 5
Electronic Location ID: e2801
Received 2016 Sep 30; Accepted 2016 Nov 18
Copyright: ©2017 Leardi et al.
Copyright year: 2017
Copyright holder: Leardi et al.
License: This is an open access article distributed under the terms of the Creative Commons Attribution License, which permits unrestricted use, distribution, reproduction and adaptation in any medium and for any purpose provided that it is properly attributed. For attribution, the original author(s), title, publication source (PeerJ) and either DOI or URL of the article must be cited.
License URL: https://creativecommons.org/licenses/by/4.0/

Keywords: Braincase, Phylogeny, Late jurassic, Micro CT, Crocodylomorpha

Funding: Foncyt PICT 2013/2725 0132, 1288 NSF EAR 0922187 1636753 Research support was provided by Foncyt PICT 2013/2725 (JML); Foncyt PICT 0132, 1288 (DP); and, NSF EAR 0922187, 1636753 (JMC). The funders had no role in study design, data collection and analysis, decision to publish, or preparation of the manuscript.

==============================
Background

Macelognathus vagans Marsh, 1884 from the Late Jurassic Morrison Fm. of Wyoming was originally described as a dinosaur by Marsh and in 1971 Ostrom suggested crocodilian affinities. In 2005, Göhlich and collaborators identified new material of this species from Colorado as a basal crocodylomorph. However, a partial skull found in association with mandibular and postcranial remains was not described.

Methods

Due to the small size and delicate structures within the braincase, micro CT studies were performed on this specimen. The new anatomical information was incorporated in a phylogenetic dataset, expanding both character and taxon sampling.

Results

This new material reinforces the non-crocodyliform crocodylomorph affinities of Macelognathusas it bears a large otic aperture, unfused frontals and lacks ornamentation on the dorsal cranial bones. The internal structures also support these affinities as this specimen bears traits (i.e., heavily pneumatized and expanded basisphenoid; the presence of additional pneumatic features on the braincase; and the otoccipital-quadrate contact) not present in most basal crocodylomorphs. Furthermore, the presence of a wide supraoccipital and a cranioquadrate passage are traits shared with Almadasuchus from the early Late Jurassic of Argentina. Macelognathus was recovered as one of the closest relatives of crocodyliforms, forming a clade (Hallopodidae) with two other Late Jurassic taxa (Almadasuchus and Hallopus).

Discussion

The clade formed by Almadasuchus + Hallopus + Macelognathus, the Hallopodidae, is characterized by a higher degree of suturing of the braincase, posteriorly closed otic aperture (paralleled in mesoeucrocodylians) and cursorial adaptations. Also, the phylogenetic position of this lineage of derived crocodylomorphs as the sister group of Crocodyliformes implies a large amount of unsampled record (ghost lineage), at least 50 million years.

Introduction

Macelognathus vagans Marsh, 1884 was originally described by Marsh (1884), and since then it has been surrounded by several controversies. The holotype specimen was recovered in 1880 at Como Bluff (Wyoming, USA), in levels belonging to the Morrison Formation. Based on the isolated mandibular symphysis that bears no teeth, Marsh (1884) assigned this taxon to a chelonian reptile. However, several authors questioned Marsh’s assignment and referred the type specimen of Macelognathus vagans to Dinosauria, either as a coelurosaur (Theropoda; Baur, 1891) or even as an ornithischian (Simpson, 1926; Von Huenne, 1956). Later, Ostrom (1971) reexamined the holotype specimen (YPM 1415), and based on the anatomical evidence reidentified Macelognathus vagans as a crocodilian.

Exploration of other outcrops of the Morrison Formation in Colorado (Brushy Basin Member; Fruita Paleontological Area) during the late 70 s and early 80 s retrieved additional material of Macelognathus. Göhlich et al. (2005) established that this material belongs to Macelognathus based on the presence of mandibular remains with the same peculiar anatomy of the type specimen: a spatulate anterior end that lacks any teeth. In that contribution, Göhlich et al. (2005) described new cranial and postcranial materials of the taxon and concluded, based on that material, that Macelognathus has non-crocodyliform crocodylomorph affinities. However, given the incomplete nature of the remains, no clear relationship among basal crocodylomorphs was claimed. Later on, Pol et al. (2013) described a new taxon from the Late Jurassic of Argentina (Almadasuchus) and noted some similarities between the femur of this new taxon and Macelognathus, but the large amount of missing data precluded the finding of a stable phylogenetic placement for the latter.

Upon personal examination (J. Leardi and D. Pol, 2012) of the collections of the Los Angeles County Museum (LACM), we were able to find a partial braincase among the material of Macelognathus retrieved from the Fruita Paleontological Area (Figs. 1A–1D). The objective of the present contribution is to describe this new material and to evaluate its phylogenetic affinities. We use high-resolution computed tomography (CT) to study the internal anatomy of Macelognathus (Figs. 2A–2E). This contribution represents the first study of CT scans from a non-crocodyliform crocodylomorph, a group whose internal braincase anatomy is highly relevant to their evolution (Clark, 1986; Walker, 1990; Wu & Chatterjee, 1993; Pol et al., 2013).

Figure 1 Posterior region of the skull of Macelognathus vagans (LACM 5572/150148).

(A–B), dorsal; and, (C–D), lateral view. Scale bar equals 1 cm. Abbreviations: boc, basioccipital; bsf, basisphenoid; cqp, cranioquadrate passage; f, frontal; lsf, laterosphenoid; ota, otic aperture; otc, otic capsule; otoc, otoccipital; p, parietal; prf, prefrontal; pro, protic; q, quadrate; soc, supraoccipital.

Figure 2 Digital reconstruction of the segmented posterior region of the skull of Macelognathus (LACM 5572/150148).

(A) dorsal; (B) ventral; (C) right lateral; (D) anterior; and, (E) posterior views.

Materials and Methods

CT analysis

The skull of Macelognathus (LACM 5572/150148) was scanned at the Microscopy and Imaging Facility of the American Museum of Natural History, using a high resolution CT scanner (GE Phoenix v|tome| × s 240). The partial skull was scanned in the transverse plane, resulting in a total of 982 slices, each with a slice thickness of 0.038 mm, with interslice spacing of 0.038, and a pixel resolution of 782 ×722. The matrix was eliminated and the individual bones were segmented using the Mimics software (V 16; Materialise, Belgium). Slice archive data is available online (http://morphobank.org/permalink/?P2550).

Systematic paleontology

Archosauria Cope, 1869	
Crocodylomorpha Hay, 1930, sensu Walker, 1970	
Hallopodidae Marsh, 1881	

Discussion: This family has been little used since its initial erection, and then only as a monotypic taxon. Here we apply this taxon to the clade found by our phylogenetic analysis comprising Hallopus victor (Marsh, 1890), Macelognathus vagans, and Almadasuchus figarii figarii Pol et al., 2013. We redefine it here as all taxa more closely related to Hallopus victor (Marsh, 1890) than to Protosuchus richardsoni Brown, 1933 or to Dibothrosuchus elaphros Simons, 1965.

Diagnosis: Hallopodidae can be diagnosed by the following synapomorphies: the presence of a cranioquadrate passage that is not in the lateral border of the skull; femoral head and distal condyles of the femur having parallel long axes; a trochanteric crest on the femur; and, a pseudointernal trochanter in the posterolateral end of the proximal end of the femur (paralleled in Kayentasuchus + Dromicosuchus). Other traits that might diagnose the clade include: greatly elongated radiale (only preserved in Almadasuchus and Hallopus, paralleled in Junggarsuchus); dorsoventrally large otic appertures (present in Almadasuchus and Macelognathus, also paralleled in Junggarsuchus), and, a quadrate-laterosphenoid contact but leaving the prootic exposed within the supratemporal fossa (only known in Almadasuchus).

Macelognathus vagans Marsh, 1884

Holotype: YPM 1415, articulated anterior portions of both dentaries, which lack teeth on the anterior end.

Referred materials: LACM 5572/150148, left dentary, partial braincase, and postcranial remains including dorsal vertebrae, a partial left ilium, partial femora, and other elements of the hindlimbs; LACM 4684/128271, partial right and left hindlimbs; LACM 4684/128272, a left femur which may belong to LACM 4684/128271; LACM 5572/150211, both calcanea, and metatarsal III; LACM 4684/133772, a portion of the right maxilla, and both dentaries. For a more detailed explanation of the elements included the reader is encouraged to refer to Göhlich et al.’s (2005) paper.

Horizon and locality: The holotype specimen was recovered from Como Bluff, Albany County, Wyoming (USA); the referred specimens were found in the Fruita Paleontological Area, Mesa County, Colorado (USA). In both of these localities the fossils derive from the Brushy Basin Member in the upper part of the Morrison Formation (Upper Jurassic, Kimmeridgian-early Tithonian; Foster, 2003).

Emended diagnosis: A non-crocodyliform crocodylomorph with dorsoventrally flattened and anteriorly edentulous dentary*, heterodont dentition, and tooth crowns devoid of mesial and distal serrations; dentary lacking caniniform teeth; maxilla with laterally concave and ventrally sinuous alveolar margin; lateral longitudinal ridge above alveolar margin of maxilla; at least two enlarged anterior maxillary teeth; enlarged maxillary teeth serrated only distally; quadrate with five pneumatic cavities; laterally closed cranioquadrate passage; large otic aperture two-thirds of the height of the quadrate; quadrate contacts the otoccipital; wide supraoccipital in posterior view; posterior bony ring on the basisphenoid, allowing the precarotid recess to open posteriorly*; internal carotids exit anteroventrally from the hypophyseal fossa*; presacral vertebrae with large neural canals (almost as large as the centrum); ilium without supraacetabular crest; ventral margin of preacetabular process of ilium thickened and medially projecting; round femoral head oriented medially and separated from proximal end by a distinct neck; proximal facet of tibia distinctly slanted laterally; longitudinal groove on proximoanterior end of fibula; calcaneum with a completely flat distal surface; strong medioplantar crest on medial base of calcaneal tuber*; overlapping proximal ends of metatarsals. Autapomorphies are marked with an asterisk (*).

Results

General features

The cranium of Macelognathus vagans (LACM 5572/150148) was first described by Göhlich et al. (2005). In this contribution, the maxilla was carefully described, but the posterior region of the braincase (mislabeled in that paper as LACM 4684/133772) was only briefly mentioned. In particular, the parietal was the only element described of the posterior specimen, highlighting two details: the presence of a weak sagittal crest and a concave nuchal crest that delimited the occipital surface of the skull. With later mechanical preparation of the specimen, the parietal was disarticulated and is no longer associated with the rest of the skull. No further details were given about the rest of the braincase of Macelognathus by Göhlich et al. (2005).

The skull of LACM 5572/150148 measures 53 mm from the occipital condyle to almost the anterior end of the frontals (Figs. 1A–1D and 2A–2E), being anteroposteriorly shorter than the length of the mandibular symphysis of the holotype specimen (YPM 1415, approximately 58 mm (Göhlich et al., 2005)). Thus LACM 5572/150148 probably belonged to a small juvenile individual, an idea further reinforced by the strongly convex skull roof (Figs. 1C–1D and 2C). This fossil preserves a partial posterior region of the skull including partial remains of the braincase. The braincase of this specimen is poorly exposed, so in order to recover more information from this region a high resolution CT scan was performed (see ‘Materials and Methods’).

Other crocodylomorph remains were recovered from the same outcrop, including crocodyliforms (Goniopholis (Göhlich et al., 2005) and Fruitachampsa (Clark, 2011)). The “sphenosuchian” affinities of LACM 5572/150148 are clearly evidenced by the very large otic aperture (Figs. 1C–1D), unlike the smaller one present in basal crocodyliforms (e.g., Protosuchus richardsoni and Fruitachampsa), and by the lack of ornamentation in the cranial roof (Figs. 1A–1B). Also, the unfused frontals might reveal non-mesoeucrocodylian affinities, but this lack of fusion might be due to an early ontogenetic stage.

Description

Both prefrontals are preserved on LACM 5572/150148, but they are only represented by the dorsal process of this bone (Figs. 3A–3D). The prefrontal contacts medially with the anterolateral edge of the frontal only. This is rare condition among crocodylomorphs, as the frontals usually contact also with the posterolateral edge of the nasals. However, no partial remains of the nasals could be identified on the anteriormost end of LACM 5572/150148, but this possibility should not be discarded as this region is badly preserved, bearing many cracks and fissures. Where this contact is better preserved, on the right side of the skull, the suture is almost straight (Fig. 3A). The prefrontals are anteroposteriorly elongate in dorsal view and they form the anteromedial border of the orbit. Anteriorly, the prefrontals are almost laminar, while posteriorly they have a higher dorsoventral development. In dorsal aspect, the prefrontal has a posterior process projected posterolaterally (Figs. 3A–3B). The posterolateral edge of the prefrontals is broadened in many basal crocodylomorphs (Pseudhesperosuchus, Dromicosuchus, Hesperosuchus agilis (CM 29894), Terrestrisuchus, Saltoposuchus, Litargosuchus, Sphenosuchus, Dibothrosuchus, Junggarsuchus), but only in Macelognathus does it form a distinct process. The CT data exposed the internal morphology of the right prefrontal. The posterior region of the right prefrontal bears a triangular pneumatic recess, with the apex pointed posterodorsally, and increases in size towards the anteroventral region of the prefrontal (Figs. 3B–3D). Ventral to this pneumatic recess a posteroventral bulge is partially preserved, which represents the descending process of the prefrontal, but its ventral end is not preserved.

Figure 3 Digital reconstructions of the frontals, prefrontals and postorbital of Macelognathus.

(A) dorsal; (B) ventral; and, (C) lateral views. (D) Close up to the right prefrontal in ventrolateral view. Same color reference as Fig. 2. Abbreviations: cc, crista cranii; ch, depression for the cerebral hemispheres; dob, depression of the olfactory bulb; fr, frontal; ifs, interfrontal suture; om, orbital margin; po, postorbital; prf, prefrontal; prfpc, pneumatic cavity of the prefrontal; prfpp, posterior process of the prefrontal; vp ifs, ventral process of the interfrontal suture.

The frontals are paired and have a clear median suture between them along their entire length (Figs. 3A–3B). The frontals are separated from each other anteriorly (where both frontals barely touch each other) but they are tightly joined to each other along the posteriormost region of the dorsal surface of the frontals. The dorsal surface of the frontals is smooth (Fig. 3A) as in most non-crocodyliform crocodylomorphs (e.g., Hesperosuchus, Sphenosuchus, Litargosuchus) with the exception of Almadasuchus, which is sculpted, and Dibothrosuchus, which has three longitudinal ridges. The frontals are rectangular in dorsal view, although their anterior end is lateromedially narrower than the posterior one, and convex in lateral view, but to a lesser extent than the parietals (see below). The frontals wedge between the prefrontals (see above), displaying a rather anteroposteriorly elongated contact. The prefrontal-frontal contact extends approximately along one third of the preserved length of the frontals. Just anterior to the mid-length of the frontals as preserved, a constriction is present on the left element (damaged on the right one) that is here interpreted as the dorsal margin of the orbits. Unlike the condition of most eusuchians and some close relatives of that clade (Glen Rose Form, Bernissartia, Shamosuchus), the dorsal margin of the orbits is not dorsally projected (Fig. 3C).

The posteroventral margins of the frontals have ventrally directed flanges, the cristae cranii (Figs. 3B–3C). These flanges contact the anterodorsal end of the laterosphenoids, as in most archosauriforms. Even though these crests are only partially preserved, they deeply project ventrally, as in most non-crocodyliform crocodylomorphs. The posteromedial region of both frontals is not preserved. Partial remains of the parietals are preserved on their posterolateral ends allowing the observation of the lateral region of the frontoparietal suture, which has its main axis anterolaterally oriented (Figs. 2A and 2C). Anterior to this region a slightly depressed area is delimited anteriorly by a curved crest. This crest is continuous with a crest present on the parietal (see below), so it is interpreted as the anterior border of the supratemporal fossa.

The ventral surface of the frontals bears two wide grooves that extend anteroposteriorly along its anterior half, interpreted here as the depression for the olfactory bulbs (Fig. 3B). These grooves are deep, limited laterally by the cristae cranii, and attenuate approximately at the level of the posterior end of the orbit. The division in two grooves is given by a well-developed and ventrally projected crest located medially at the interfrontal suture. This crest would imply a partially divided (or bilobed) olfactory bulb, and is much more developed than in crocodyliforms (e.g., Araripesuchus gomesii Price, 1959 (AMNH 24450); Sebecus icaeorhinus Simpson, 1937; Crocodylus acutus Cuvier, 1807 (Colbert, 1946)). The frontals of Macelognathus also have two elongated depressions on the posterior half of the ventral surface of the frontals, which are interpreted here as the impressions of the cerebral hemispheres (Fig. 3B). These depressions have a considerable anteroposterior development, occupying 25% of the total length of the frontal.

A fragment of bone is preserved articulated to right frontal along its mid-length, interpreted as part of the postorbital (Figs. 3A–3C). The region preserved corresponds to the postorbital plate, just anterior to the descending process.

The parietals are badly damaged, with almost the whole left part of the bone missing (Figs. 4A–4B). The almost complete lack of the left half of the bone precludes determining if the parietals were fused. The central right half of the parietal was already described previously (Göhlich et al., 2005; see above), and for additional information on that region the reader is referred to that contribution. Thus, most of the description of the parietal will be based on the right half of this element. Most crocodylomorphs have the parietals fused as a single element, with the exception of Hesperosuchus, Pseudhesperosuchus, Dromicosuchus, Terrestrisuchus and Saltoposuchus (Clark, Sues & Berman, 2000); although in some cases this could be due to the early ontogenetic stage of the specimens. The parietal is strongly convex in lateral view, a condition that is more evident at its contact with the supraoccipital (Figs. 1C–1D and 2C–2E). The dorsal surface of the parietal is smooth, as in most basal crocodylomorphs (Hesperosuchus, Pseudhesperosuchus, Dromicosuchus, Terrestrisuchus, Saltoposuchus, Litargosuchus, Kayentasuchus, Sphenosuchus, Dibothrosuchus, Junggarsuchus and Almadasuchus), contrasting with the ornamented condition present in crocodyliforms (e.g., Protosuchus richardsoni Brown, 1933).

Figure 4 Digital reconstructions of the parietal of Macelognahus.

(A) dorsal; and, (B) lateral views. Abbreviations: alpf, anterolateral process of the frontal; dc, dorsal crest; f. lsf, facet for the laterosphenoid; f. pro, facet for the protic; STF, border of the supretemporal fossa.

Anteriorly, only the lateral regions of the sutures of the parietals with the frontals were preserved. This anterolateral border is oriented obliquely, as described above, with the lateralmost end projected more anteriorly than the medial one. At the midpoint of this suture an acute anterior process is present in both parietals. This process bears a well-developed anteroposteriorly oriented crest on its dorsal surface (Fig. 4B). The dorsal crest of the anterior acute process of the parietal is anteroposteriorly short, disappearing at approximately one quarter of the total length of the parietal. Lateral to this crest the surface of the parietal bears a concavity that occupies almost two thirds of the total length of the bone. This region is identified as the supratemporal fossa. Its great anteroposterior development is consistent with the condition of most basal crocodylomorphs, in which the supratemporal fossae are very large and occupy almost the entire dorsal surface of the parietal.

Along its ventral edge, the parietal contacts the laterosphenoid and the prootic. The contact with the former is extended through the anterior 75% of the ventral margin of the parietal along a straight suture (Fig. 4B). The posterior fourth of the ventral border of the parietal contacts the prootic. Posteriorly, the parietal is very fragmentary, thus its contact with the supraoccipital is not preserved. The internal surface of the parietal is concave, which is especially marked on the posterolateral region of the medial surface of the parietal, near the sutural contact with the laterosphenoids. This depressed internal surface, which continues on to the laterosphenoids, is interpreted as the posterolateral expansion of the cerebral hemispheres.

The quadrate, despite being damaged in many regions, is one of the most complete bones of the skull. Both quadrates are preserved in LACM 5572/150148, although the right one is better preserved than the left one, as with most elements of the skull (Figs. 2A–2E). The quadrate body is curved posteriorly in lateral view, being markedly convex anteriorly. One of the most notable features of the right quadrate is the presence of a large otic aperture on its main body (Figs. 5A–5B). The quadrate only forms the ventral and anterior borders of this fenestra. Despite being incomplete (as the squamosal is not preserved) we can infer its size, which is approximately two thirds of the dorsoventral length of the quadrate. The presence of a large otic aperture, where the quadrate forms the anteroventral borders of a well-delimited fenestra, is a feature shared with Almadasuchus and Junggarsuchus. In these taxa the otic aperture is also dorsoventraly large, attaining at least half of the height of the quadrate body. In other known non-crocodyliform crocodylomorphs (Hesperosuchus “agilis,” Pseudhesperosuchus, Dromicosuchus, Terrestrisuchus, Sphenosuchus, Dibothrosuchus), the otic aperture is not such a circumscribed fenestra as the otic aperture lies posterior to the quadrate body and it is not closed posteriorly (see below). However, this condition is unknown in Kayentasuchus and Litargosuchus as the posteroventral region of their quadrates are not well preserved.

Figure 5 Digital reconstructions of the right quadrate of Macelognathus.

(A) lateral; (B) medial; (C) anteromedial; and, (D) posterior views. Abbreviations: cqp, cranioquadrate passage; dmq, dorsomedial process of the quadrate; f., qj, facet for the quadratojugal; f. otoc, facet for the otoccipital; lcq, lateral condyle of the quadrate; mcq, medial condyle of the quadrate; oj, otic joint; ota, otic aperture; ptq, pterygoid ramus of the quadrate; qf, quadrate fenestra; tc, tympanic crest; 1–5, internal chambers of the quadrate.

Anterior to the otic aperture, on the posterior surface of the quadrate body, a crest is present (Fig. 5D). Given its anterior position to the otic aperture, this crest is here identified as the tympanic crest. The lateral surface of the main body of the quadrate of Macelognathus is slightly concave and this concavity is limited anteriorly by an oblique oriented crest. On the anterolateral border of the body of the quadrate an elongated and obliquely oriented facet can be seen. This facet is for the articulation of the quadratojugal on the quadrate (Fig. 5A). The precise dorsal extent of this facet is unknown, but it seems not to reach the dorsalmost end of the quadrate anterolateral surface. Thus, the quadrate participates in the posterodorsal border of the infratemporal fenestra as in most non-crocodyliform crocodylomorphs (e.g., Dromicosuchus, Dibothrosuchus, Sphenosuchus, Almadasuchus). In crocodyliforms (e.g., P. richardsoni, Edentosuchus) the quadrate is excluded from the margin of the infratemporal fenestra, as the quadratojugal covers all the anterior surface of the quadrate that could be exposed in this fenestra.

Also on the anterior margin of the quadrate body, and dorsal to the facet for the quadratojugal the posterior border of an elongated rounded fenestra is present (Figs. 5A and 5C). This quadrate fenestra is located at the dorsoventral midpoint of the anterolateral surface of the quadrate and appears to be pneumatic, as it connects with the internal space of this bone (see below). Among non-crocodyliform crocodylomorphs, pneumatic quadrate fenestrae are also present in simple form in Terrestrisuchus and more complexly in Junggarsuchus and Almadasuchus. These pneumatic fenestrae are always located on the main body of the quadrate and anterior to the tympanic crest. However, there is a great deal of variation in the shape and size of these fenestrae among different taxa. The borders of the quadrate fenestra of Terrestrisuchus have at least some participation with the quadratojugal on the anterior end (Crush, 1984), while in Dibothrosuchus, Junggarsuchus, Almadasuchus and crocodyliforms (e.g., Protosuchus richardsoni, Gobiosuchus) the quadrate fenestrae are completely surrounded by the quadrate. As was previously mentioned, this condition cannot be precisely known in Macelognathus due to the incomplete preservation of the quadrate fenestra. Furthermore, these fenestrae are variable in size and in most taxa are small, while the quadrate fenestrae of Terrestrisuchus and Almadasuchus are elongated and quite large, being at least half of the dorsoventral height of the quadrate body. Junggarsuchus is unusual also in this aspect, as at least two fenestrae are identified on its quadrate (M Klein et al., 2016, unpublished data). This condition is similar to the one present in basal crocodyliforms which have multiple quadrate fenestrae (e.g., Protosuchus richardsoni). Finally, the condition of Dibothrosuchus deserves mention. This structure was not described in the original paper (Wu & Chatterjee, 1993), but upon personal examination (D. Pol), we identified what could be interpreted as the posterior border of a large quadrate fenestra on the left quadrate. The anterolateral margin of the quadrate of the skull of the referred specimen (IVPP V 7907) bears an elongate, rounded border on its anterior end, and thus can be interpreted as the posterior border of a pneumatic quadrate fenestra. The quadratojugal is very incompletely preserved and its relationships with neighboring elements are unclear (Wu & Chatterjee, 1993).

The orbital process of the quadrate is preserved on the right quadrate on its dorsomedial region (Figs. 5B–5D). The orbital process of the quadrate is a lateromedially thin and medially projected process that contacts the dorsolateral region of the prootic. The articular surface for the prootic is exposed only medially (or slightly dorsomedially). The quadrate-prootic contact is anteroposteriorly long, extending through almost the entire length of the prootic as in most crocodylomorphs (Fig. 2C). The dorsolateral region of the quadrate is not preserved in any of the elements, precluding observation of the primary head of the quadrate.

Medially, on the lateral surface of the adductor chamber, the anterior surface of the quadrate is slightly convex and mediolaterally wide (Fig. 5C). Although this region is badly preserved, no crest can be observed on the anterior surface of the quadrate, just anterodorsal to the quadrate condyles. A broad and relatively short pterygoid ramus of the quadrate is present in the ventromedial region of the adductor chamber. The distal end of the pterygoid ramus of the quadrate is asymmetrical, being more expanded dorsally. The ventral region of the pterygoid ramus of the quadrate seems to be separated from the most dorsal part by an anteromedial notch. A similar condition is present in Almadasuchus, where the quadrate is greatly expanded medially and this notch represents the lateral border of the passage for the middle cerebral vein (sensu Walker, 1990). The distal end of the left quadrate bears well-preserved distal condyles, which are divided by a shallow intercondylar groove. The lateral condyle is mediolaterally broader than the medial one, while the latter is slightly more distally projected.

The body of the quadrate and the distal condyles are heavily pneumatized, being hollow internally with thin walls. The body of the quadrate is internally divided from dorsal to ventral in a series of individualized interconnected chambers. A total of five (5) internal divisions, which increase in volume towards the distal condyles, can be identified in the quadrate (Fig. 5C). The first chamber, located on the most dorsal region of the quadrate is approximately triangular in dorsal view. The exact size of this chamber cannot be precisely known, as the dorsal region of the quadrate is incomplete. The dorsal chamber is separated from the second one by a thin bony septum only developed on the posterior half of the inner cavity of the quadrate. The second chamber is parallelogram shaped in dorsal view and is dorsoventrally higher than anteroposteriorly long. The second chamber communicates with the central chamber via several passages on its ventral septum. One is located on the posterior border (the largest one) and three on the anteromedial border (the medial most is the largest of these). The third and central cavity is volumetrically smaller than the second, being dorsoventrally flat when it is compared with the other internal divisions of the quadrate. The bony septum dividing the central chamber of the fourth chamber seems to be complete, whereas the incompleteness on the anterolateral region could be attributed to preservational features. The communication with the ventral chamber is across a small foramen located on the anteromedial border of the chamber. The fourth chamber is dorsoventrally higher than the third, even though it is lateromedially wider than its dorsoventral development. The posteromedial border appears to be open, as it is confluent with the cavity formed within the quadrate and limited externally by the otic aperture (middle ear). This fourth chamber is also externally connected via the pneumatic quadrate foramen described above (usually called the quadrate fenestra in most crocodylomorphs). The fourth chamber is separated from the ventralmost one by an incomplete septum which has three communicating foramina, one posteriorly and two anteriorly. Finally, the largest pneumatic chamber in the quadrate is the ventral one. This chamber occupies the whole interior volume of the distal body of the quadrate, including the pterygoid process of the quadrate. Pneumaticity invades the distal condyles via three lateral diverticula. The ventral chamber is open ventrally as in other derived non-crocodyliform crocodylomorphs (e.g., Almadasuchus and Dibothrosuchus).

On the posterodorsal region of the quadrate a large, pointed and dorsomedially directed process is present (Fig. 5D). This process forms the posterior border of the otic aperture and contacts through its medial surface the ventrolateral margin of the otoccipital. The quadrate-otoccipital contact is also present in Junggarsuchus, Almadasuchus and crocodyliforms, but not in more basal “sphenosuchians” (Pol et al., 2013). However, the morphology of this region of the quadrate is unknown in some of the other non-crocodyliform crocodylomorphs. The dorsal surface of the posterodorsal process of the quadrate of Macelognathus bears a dorsomedially oriented furrow (Figs. 5A and 5D). The position and direction of this groove and the fact that the quadrate forms its ventral border are consistent with the cranioquadrate passage. Such passages are absent in most “sphenosuchians” and even basal crocodyliforms (Protosuchus richardsoni and Orthosuchus). On the other hand, cranioquadrate passages are present in Almadasuchus and in the vast majority of crocodyliforms (such as Gobiosuchus, Sichuanosuchus and all mesoeucrocodylians).

The prootic, although damaged, is preserved almost complete on the right side of the skull (Figs. 2A–2E and 6A–6B). The left prootic only preserves part of its ventral region and the ventrolateral flange (see below). The protic is a dorsoventrally high bone when it is compared with its lateromedial development (Figs. 7A–7D). As in most non-crocodyliform crocodylomorphs (e.g., Hesperosuchus, Sphenosuchus, Dibothrosuchus, Almadasuchus) and thalattosuchians (e.g., Cricosaurus araucanensis (Gasparini & Dellapé, 1976)) the prootic is exposed on the posterodorsal region of the supratemporal fossa. In Macelognathus the prootic is verticalized on the supratemporal fossa, with the exposed surface facing anteriorly, a similar condition to the one observed in Dibothrosuchus, Sphenosuchus and Kayentasuchus. On the other hand, the prootic is exposed facing anterodorsally within the supratemporal fossa in Almadasuchus, Litargosuchus and thalattosuchians. In dorsal view, the prootic contacts the squamosal posteriorly (not preserved) and anteriorly the parietal along its medial margin.

Figure 6 Digital reconstructions of the braincase of Macelognathus.

(A) lateral; and, (B) medial views. Same color reference as Fig. 2. Abbreviations: aur, auricular recess; bot, basal tubera; bsf ros, rostrum of the basisphenoid; cif, crista interfenestralis; coc, cochlear recess; con, occipital condyle; cpr, crista prootica; fov, fenestra ovalis; g. csc, groove for the caudal semicircular canal; ma, mastoid antrum; met, metotic fissure; otc, otic capsule; PTR, posterior tympanic recess; paroc, paroccipital processes; vp bsf, ventral plate of the basisphenoid.

Figure 7 Digital reconstructions of the right protic of Macelognathus.

(A) lateral; (B) medial; (C) dorsal; and, (D) ventral views. Abbreviations: aur, auricular recess; coc, cochlear recess; cpr, crista prootica; f. boc., facet for the basioccipital; fma, foramina of the lateral wall of the mastoid antrum; f. otoc, facet for the otoccipital; f. q, facet for the quadrate; fov, fenestra ovalis; lsc, groove for the lateral semicircular canal; ma, mastoid antrum; VII, exit for the VII cranial nerve.

The dorsal region of the prootic is more lateromedially developed, and houses a large mastoid antrum (Figs. 7A and 7C). This pneumatic cavity is subrectangular in shape and is restricted to the prootic, although the squamosal (not preserved) might form the dorsal roof of the mastoid antrum. The mastoid antrum appears to communicate with the middle ear via at least two large foramina on the dorsolateral surface of the prootic: a larger one located on the anterior half and a smaller, more elongated one, located on the posterior half of this surface (Figs. 6C–6D). One or more ventral openings could be present on the anteroventral surface of the mastoid antrum, but these may have been caused by breakage of the specimen. Also, the presence of such ventral apertures of the mastoid antrum would depend on the particular shape of the unpreserved contact with the pterygoids. Multiple lateral openings of the mastoid antrum into the middle ear cavity are present in Kayentasuchus, Dibothrosuchus and P. haughtoni; while in Sphenosuchus the mastoid antrum opens through a unique undivided foramen.

Anteroventral to the mastoid antrum the prootic is damaged, preserving the articulation with the laterosphenoid only anteriorly. Thus, the structure of the trigeminal recess is unknown. The anteroventral region of the prootic has many discontinuities on its lateral wall, but interpreting these as evidence of the exit of cranial nerves would be equivocal. However, in the central region of this area, ventral to the mastoid antrum, and near the posterior edge of the prootic a rounded foramen is present within the prootic (Figs. 7A–7B). The placement of this foramen, anteroventral to the fenestra ovalis (see below), and its orientation indicates it is the exit of the facial nerve (VII). The structure of this foramen closely resembles the one present in Protosuchus haughtoni (Busbey III & Gow, 1984) as it is dorsoventrally oriented, while in Sphenosuchus (Walker, 1990) an oblique groove is associated with the exit of the VII. Posterodorsal to this area and ventral to the posterolateral opening of the mastoid antrum of Macelognathus is a small anteroposteriorly oriented foramen that continues posteriorly as a groove. The groove associated with the foramen continues its straight path onto the otoccipital. The foramen continues anteriorly as a passage to reach an expanded cavity, located dorsal to the exit of the VIIth nerve. This configuration is consistent with its assignment as the groove and bony canal of the lateral semicircular canal of the inner ear that has been exposed laterally by breakage of the specimen; while the enlarged cavity located anterior to it is interpreted as the ampulla of the lateral and rostral semicircular canals. Furthermore, from the anterodorsal end of the expanded cavity (ampulla) a dorsally directed canal arises and continues its path dorsally, just medial to the mastoid antrum. Upon reaching the anterodorsal end of the prootic, it turns posteriorly very abruptly, leaving a groove on the dorsal surface of the prootic. This groove continues posteriorly on the medial aspect of the supraoccipital (see below). These traits are consistent with the identification of this structure as the rostral semicircular canal.

Ventral to the region of the mastoid antrum and the groove for the lateral semicircular canal, in the central region of the posterior margin of the prootic, a rounded notch is present (Fig. 7A). The shape of this notch is due to the lack of preservation of the anteriormost area of the otoccipital. The position and the bones that form this notch (i.e., the prootic forms its anterior and dorsal margins) help us to identify it as the fenestra ovalis. Ventral to the fenestra ovalis, the prootic bears a large laterally projected flange (prootic-basisphenoid flange sensu Walker (1990) or crista prootica). This flange extends to the ventral margin of the prootic and continues also on the dorsal region of the basisphenoid, as in Sphenosuchus. The otoccipital contacts the prootic-basisphenoid flange along the posterior margin of its prootic part. The prootic-otoccipital contact also involves the posterior surface of the prootic (excluding the region of the fenestra ovalis), although the central region of the prootic is not preserved. This contact is preserved on the dorsal and ventral region of the posterior surface of the prootic. This double contact involving the otoccipital and the prootic encloses a dorsoventrally oriented pyramidal cavity interpreted as the cochlear recess (=lagena). Further ventrally the prootic contacts the basisphenoid through its posteroventral margin, and as a result closes the cochlear recess ventrally.

The medial surface of the prootic is damaged, exposing parts of the bony labyrinth medially. Dorsal to the bony canal of the lateral semicircular canal and its associated ampulla a strongly projected medial bony flange is present. This flange is sigmoidal in medial view and it limits ventrally a deep depression. This depression is capped dorsally by the supraoccipital. A similar depression has been observed in medial view of the braincase of Sphenosuchus and has been interpreted as the auricular (or floccular) recess of the cerebellum (Walker, 1990) (Figs. 6A and 7B). Medial views of the braincases of other basal crocodylomorphs are rare, precluding the observation of this feature.

The opisthotics and exoccipitals are fused constituting an otoccipital. The right otoccipital preserves part of the subcapsular buttress, the otic capsule, the distalmost part of the ventrolateral part of the parocipital process and the dorsal region of the bone (just dorsally to the otic capsule) (Figs. 6A–6B). The subcapsular buttress of the otoccipital is a rounded eminence limiting the posteroventral region of the otic capsule forming the lateral and posterior walls of the cochlear recess (Figs. 8A–8B). The subcapsular buttress of the otoccipital contacts the crista prootica anteriorly to form the crista interfenestralis, which is only preserved at its base on this specimen. The posterior border of the subcapsular buttress represents the ventral border of the metotic fissure, from where the cranial nerves IX–XI exit the endocranial cavity. Internally, this region is divided in two elongated slits by a lamina of the otic capsule, also formed by the otoccipital (Fig. 8B). The dorsal slit, located at the posterior border of the metotic fissure is interpreted as the exits of the cranial nerves X–XI and the posterior cerebral vein; while the ventral slit, located at the ventral border of the metotic fissure, is interpreted as the exit of cranial nerve IX. A similar division is also present in Sphenosuchus (Walker, 1990: Fig. 24), although this region of the braincase is not that well exposed in other crocodylomorphs. On the dorsal border of the otic capsule a posteriorly directed groove is observed, the extension of the groove conducting the lateral semicircular canal observed in the prootic (Fig. 8C). This groove continues medially up to the level of the otic capsule, and at the posterior border of the capsule it expands into the ampulla of the caudal semicircular canal. Dorsal to this region an oblique groove runs through the dorsomedial surface of the otoccipital, medial to mastoid antrum. This groove continues anteriorly on the ventral surface of the supraoccipital and represents the caudal semicircular canal.

Figure 8 Digital reconstructions of the right otoccipital (excluding the paroccipital processes) of Macelognathus.

(A) posterior; (B) lateral; and, (C) anterior views. Abbreviations: cif, crista interfenestralis; coc, cochlear recess; dl otc, dorsal lamina of the otic capsule; f. boc, facet for the basioccipital; f. pro, facet for the protic; lsc, groove for the lateral semicircular canal; met, metotic fissure; pcv, posterior cerebral vein; PTR, posterior tympanic recess; scb, subcapsular buttress; IX–X, exit for cranial nerves IX…X.

Dorsal to the otic capsule, the otoccipital is heavily pneumatized (Figs. 8A–8B). This pneumatic cavity is completely separated from the mastoid antrum by a bony wall formed by the posterodorsal region of the prootic. The otoccipital also contributes to this separation, as a thin bony lamina that articulates anterodorsally with posterior bony septum of the mastoid antrum formed by the prootic. The position of this pneumatic cavity, located dorsal to the otic capsule and to the metotic fissure, is consistent with its identification as the posterior tympanic recess (PTR sensu Wu & Chatterjee, 1993). A slight posterior tympanic recess is present in Sphenosuchus, being represented by a shallow groove between the prootic-opisthotic suture. A heavily pneumatized cavity in this same area is present in Dibothrosuchus and P. haughtoni, thus sharing the same condition with Macelognathus. The posterior tympanic recess of Macelognathus is partially divided anteriorly by a bony pillar located near the suture with the prootic. However, due to the almost complete lack of preservation of the paroccipital processes we are unable to evaluate if this pneumatic cavity penetrates these processes as in Dibothrosuchus and P. haughtoni.

Ventromedially the otoccipital contacts the basioccipital on the lateral region of the floor of the endocranial cavity. Posteriorly, the otoccipital has a medially projected process that excludes the basioccipital from the ventrolateral border of the foramen magnum. Therefore, the otoccipital forms the entire lateral margin of this foramen (Fig. 6A). The otoccipital-basioccipital contact reaches the occipital condyle, and as result the otoccipital participates in the dorsolateral margin of it. Finally, only the ventralmost part of the right paroccipital process is preserved. The fragment preserved is exposed posteroventrally and separates the basioccipital from the quadrate in this view. This region of the paroccipital process contacts with the quadrate laterally; anteriorly and medially with the basioccipital.

The supraoccipital is present between the foramen magnum and the parietals. Only the right half of the supraoccipital is preserved (Figs. 2A–2E and 6A–6B). The preserved half is rectangular in external view (Figs. 9A–9B), thus the full element would have a quadrangular shape in occipital view. This condition is similar to the one present in Litargosuchus, Junggarsuchus, Almadasuchus and crocodyliforms that have a wide supraoccipital, unlike the dorsoventrally elongated supraoccipitals of other crocodylomorphs (Pseudhesperosuchus, Terrestrisuchus, Kayentasuchus, Sphenosuchus, Dibothrosuchus). Anteroventrally, the supraoccipital bears a ventral bulge that is pyramidal shaped and with its apex oriented ventrally, giving this region a higher dorsoventral development than the rest of the supraoccipital (Fig. 9C). The anteromedial region of the anteroventral bulge bears a very deep concavity, which is the part of the auricular (or floccular) recess that extends into the supraoccipital. Dorsal to this recess a shallow groove can be observed. This groove continues on the dorsal border of the anteroventral bulge, just ventral to the dorsal plate of the supraoccipital. As mentioned above, this groove is interpreted as for the caudal semicircular canal.

Figure 9 Digital reconstructions of the supraoccipital of Macelognathus.

(A) posterior; (B) dorsal; and, (C) medial view. Abbreviations: aur, auricular recess; avb, anteroventral bulge of the supraoccipital; csc, groove for the caudal semicircular canal.

As in all “sphenosuchians” where this condition has been properly explored, the supraoccipital of Macelognathus does not participate in any pneumatic cavity (i.e., transverse canal communicating the mastoid antra from both sides) (Figs. 6A–6B), unlike the condition seen in crocodyliforms (e.g., P. richardsoni (Clark, 1986), P. haugthoni (Busbey III & Gow, 1984)) with the exception of thalattosuchians (e.g., Pelagosaurus (Pierce, Williams & Benson, 2016); Steneosaurus (Brusatte et al., 2016)). In previous contributions, a transverse canal communicating both mastoid antra was mentioned to be present in Dibothrosuchus (Wu & Chatterjee, 1993), however this condition cannot be observed in the external view of the skull material known for this taxon (IVPP V 7907; J Clark, pers. obs., 2014). Posterolaterally, the supraoccipital articulates with the otoccipital on its posterior third through an oblique interdigitated suture (Fig. 2A). Anterolaterally, the supraoccipital contacts the prootic on its anterior two thirds through an interdigitated suture on the anteroventral bulge.

The basioccipital forms the floor of the posterior half of the endocranial cavity, while the anterior half is formed by the basisphenoid (Figs. 6A–6B). Anteriorly, the median region of the basioccipital is divided by a short posterodorsal process of the basisphenoid. Lateral to this area, the basioccipital-basisphenoid articulation is straight, through a transverse suture (Figs. 10A–10C). On the anterolateral corner of the endocranial cavity, the basioccipital bears a posteroventrally elongated and obliquely oriented facet for the prootic (Figs. 10B–10C). Posterior to this facet, and along its dorsolateral surface, the basioccipital contacts the subcapsular buttress of the otoccipital. On the central region of the dorsal aspect of the basioccipital, this element has lateral expansions that contact the medial wall of the subcapsular buttress and the otic capsule. These lateral flanges contract before reaching the occipital condyle.

Figure 10 Digital reconstructions of the basioccipital of Macelognathus.

(A) ventral; (B) lateral; (C) dorsal; and, (D) posterior views. Abbreviations: bor, basioccipital recess; bot, basal tubera; bt, blind tubes of the basioccipital; cc bor, central crest of the basioccipital recess; coc, cochlear recess; con, occipital condyle; f end, floor of the endocranial cavity; f. otoc, facet for the otoccipital; f. pro, facet for the protic; lf boc, lateral flanges of the basioccipital; ml, medial lamina of the basal tubera; rhom, rhomboidal sinus; vf boc, ventral flanges of the basioccipital.

The occipital condyle of the basioccipital is rugose and has the articular surface for the atlas-axis complex oriented posteriorly, but slightly inclined dorsally. The occipital condyle is projected ventrally, delimiting a ventral rim around this area (Figs. 10B and 10D). On the ventral surface of the basioccipital, anterior to the occipital condyle, the basioccipital tubera are partially preserved. These are laminar as in most crocodylomorphs (Nesbitt, 2011) and are separated from the occipital condyle by a shallow groove (Fig. 10A). Only the right basioccipital tuber and partial remains of the medial lamina uniting both tubera are preserved. A wide depressed area is present on the ventral surface of the basioccipital, the basioccipital recess. This wide recess is present in almost all non-crocodyliform crocodylomorphs (e.g., Hesperosuchus, Dromicosuchus, Sphenosuchus, Dibothrosuchus, Almadasuchus), but is absent in Junggarsuchus and crocodyliforms (e.g., P. richardsoni, P. haughtoni, Orthosuchus). A wide crest projects anteriorly from the anteroventral surface of the medial lamina joining both basioccipital tubera throughout all the ventral surface of the basioccipital, dividing the basioccipital recess. Two posteriorly directed blind tubes excavate the posterior region of the basioccipital recess. These additional depressions on the posterior surface of the basioccipital are also present in Almadasuchus and Dibothrosuchus, while they are lacking in other basal crocodylomorphs. Anteriorly, the basioccipital bears two ventral flanges that project posterolaterally. The anterodorsal surface of these flanges is deeply excavated by two parallel rounded depressions (Figs. 10B–10C). These depressions do not perforate the basioccipital, and are limited anteriorly by the basisphenoid, which articulates with this region of the basioccipital. These expanded cavities are located posteroventral to the cochlear recess, in a similar fashion to the placement of the rhomboidal sinus (Colbert, 1946; Walker, 1990). Thus, we interpret them as such. Despite that the posteroventral region of the basisphenoid is not preserved (see below), the rhomboidal sinus seems to open directly to the pharynx. Given this morphology, we can conclude that Macelognathus lacks well-developed bony passages for Eustachian tubes, which are present in Crocodyliformes (e.g., P. richardsoni (Clark, 1986)).

In addition to the dorsal contact with the otoccipital along the endocranial cavity, the basioccipital has a contact with the otoccipital. This contact is between the distal end of the anteroventral flanges of the basioccipital and the paroccipital process (Fig. 10A).

The basisphenoid, although damaged and incomplete ventrally, is fairly well preserved. The basisphenoid is greatly expanded lateromedially in ventral view (Fig. 2B) as in Junggarsuchus, Almadasuchus and basal crocodyliforms (e.g., P. richardsoni), unlike the more compressed basisphenoids present in Terrestrisuchus, Sphenosuchus and Dibothrosuchus. CT data of the central region of the basisphenoid allowed the observation of the highly pneumatized nature of this region (Figs. 11A–11D), a condition also shared with Almadasuchus, Junggarsuchus and Crocodyliformes.

Figure 11 Digital reconstructions of the basisphenoid of Macelognathus.

(A) lateral; (B) dorsal; (C) anterior; and, (D) posterior views. Abbreviations: bsfp, basisphenoid plate; bsf ros, rostrum of the basisphenoid; car, passage for the carotid arteries; coc, cochlear recess; cp, carotid pillar; ds, dorsum sellae; f. boc, facet for the basioccipital; f. lsf, facet for the laterosphenoid; f. pro, facet for the protic; pal; passage for the palatine artery; pp bsf, posterior process of the basisphenoid; ppd, posterior depression of the basisphenoid; pr bsf, posterior “ring” of the basisphenoid; st, sella turica; VI, exit for the VI cranial nerve.

Internally, the basisphenoid is a very complex bone formed by a series of bony struts and laminae that enclose various recesses and pneumatic cavities. The posterior surface of the body of the basisphenoid is subrectangular in posterior view, with a ventral constriction present on its center (Figs. 10A–10B and 10D). This constriction is caused by a pointed, posteriorly projected process of the basisphenoid that wedges into the anterodorsal region of the basioccipital (see basioccipital above). Lateral to this process there is a flat, dorsally exposed area on each side of this median process that represents the articular facet for the basioccipital. However, the basioccipital does not contact this whole surface, as it leaves the medialmost region exposed posteriorly. These free surfaces cap the parallel rounded depressions of the basioccipital (see above). The basioccipital also contacts the basisphenoid on its ventrolateral margin, as the ventrolateral flanges of the basioccipital articulate on this area. The facet for these flanges is very elongated, almost ridge-like. The dorsolateral region of the posterior edge of the basisphenoid articulates with the prootic through a dorsally exposed facet (Figs. 11A–11B). Thus, there is a triple contact involving the basioccipital, basisphenoid, and the prootic (Fig. 6D). As was previously mentioned, this triple contact closes the ventralmost end of the cochlear recess (see prootic). Additional participation in this contact by the otoccipital is possible, as the ventrolateral region of the cochlear recess is not preserved.

Anteriorly, the basisphenoid of Macelognathus is “W”-shaped, comprising a dorsal basisphenoid plate and a central carotid pillar, which conveyed the internal carotids (Fig. 11C). The carotid pillar can be divided into the dorsal postcarotid recess and the ventral precarotid recess (Walker, 1990). The basisphenoid plate is directed anterodorsally and continues anteriorly up to the level of the hypophyseal fossa (Fig. 11A). Due to the fragmentary preservation of LACM 5572/150148 it is not possible to establish if the basisphenoid plate formed a continuous bony sheet. On the anterolateral ends of the basisphenoid plate, two elongated oblique facets for the laterosphenoids are present (Fig. 11B). Ventromedial to these facets two anterolaterally directed grooves, which have an associated foramen (only preserved on the left side) can be seen. These represent the exits for the abducens cranial nerve (VI).

Just medial to the grooves for the abducens nerve, the basisphenoid plate forms the dorsum sellae, the dorsal roof of an anteriorly placed hypophyseal fossa (Figs. 11B–11C). The hypophyseal fossa is not bounded anteriorly, as in other basal crocodylomorphs (e.g., Sphenosuchus, Almadasuchus) and crocodyliforms. Although not well-preserved (only on the right side), bulges are present lateral to the hypophyseal fossa. Anteroventrally an anteriorly exposed bony plate is present that is pierced by two foramina, the exit for the internal carotid arteries (Fig. 11C). This condition is very unusual among crocodylomorphs, as in taxa where this trait is known the exit of the internal carotids is located on the ventrolateral surface of the hypophyseal fossa (e.g., Sphenosuchus, Dibothrosuchus, P. haughtoni). The bony plate bears a dorsal bulge and, separated from it, the base of the basisphenoid rostrum is preserved. Thus the base of the basisphenoid rostrum is located well ventral to the base of the hypophyseal fossa, as in Sphenosuchus, contrasting with the condition present in Almadasuchus and Crocodyliformes (e.g., P. haughtoni, Caiman yacare Daudin, 1802). In these taxa the hypophyseal fossa is bounded anteroventrally by the basisphenoid rostrum, and in crocodyliforms this anterior process expands dorsally forming the anterior wall of the fossa. On the posterior surface of the anteriorly exposed bony plate and ventral to the hypophyseal fossa, an elliptical, lateromedially elongated recess is present. A similar rostral recess is also present in Sphenosuchus, but the structure of this region is not known in other basal crocodylomorphs.

Ventral to the basisphenoid plate two lateral bony struts project ventrolaterally. Medial to these struts, both carotid pillars are preserved on their posterior region (Fig. 11C). These have an internal hollow structure for the passage of the internal carotids. Unlike the condition present in Sphenosuchus, the carotid pillars completely enclose the internal carotids, at least in their posterior part, while in the latter taxon they run dorsally on the carotid pillar leaving a dorsally open groove (Walker, 1990). Dorsal to the carotid pillar, the postcarotid recess is only partially preserved. Posteriorly this recess bears two parallel posteriorly projected depressions that excavate this surface. Ventral to the carotid pillar the precarotid recess is also partially preserved on its posterior region where it has a bony ring delimiting a circular passage (Fig. 11D). Thus, the precarotid recess seems to be posteriorly open, unlike the condition in Sphenosuchus where this recess is closed.

Two anterolaterally oriented bony struts arise from the ventral surface of the posterior region of the basisphenoid, ventrolateral to the carotid pillars (Fig. 11A). The lateral surface of the bony struts bears a foramen that divides in two rami, an anteriorly directed ramus and an anteroventrally oriented one. The anterior ramus of this foramen communicates with the carotid pillar, representing the initial passage where the internal carotid enters the basisphenoid. The anteroventral ramus is only formed by a dorsal bony bridge and then continues anteroventrally through a slight groove. The path described by the initial part of the foramina, and then by the groove, is consistent with the identification of this structure as representing the branching of the internal carotid into the palatine artery. The two lateroventral bony struts branch out rapidly towards the ventral surface of the basisphenoid, within the highly pneumatized ventral region of this bone. These branches expand distally to form the ventrally expanded plate of the basisphenoid.

Posteroventral to the posterior edge of the basisphenoid are two isolated bony laminae (Figs. 2B and 2D). These posterolateral laminae contact the posteromedial process of the quadrate and the paroccipital processes of the ottocipital. Given their topological position these are interpreted as fragmentary remains of the pterygoids, particularly the quadrate ramus of that bone.

Although elements from both sides are preserved, the laterosphenoids are poorly preserved (Fig. 2). The laterosphenoids articulate posteriorly with the prootics through a straight overlapping suture (Fig. 12A). Posteroventrally the laterosphenoid contacts the basisphenoid dorsolateral to the hypophyseal fossa via an elongated facet (see basisphenoid above) (Fig. 12C). The region in between these sutures is missing, precluding evaluating the shape of the trigeminal foramen. Most of the dorsal edge of the laterosphenoid contacts the parietal through a straight suture. The anterodorsal end of the laterosphenoid articulates with the frontals, having short capitate processes located on its anteriormost end (only preserved on the right side).

Figure 12 Digital reconstructions of the left and right laterosphenoids of Macelognathus.

(A) lateral; and, (C) ventral views. Left laterosphenoid in (A) lateral; and, (B) posterior view. Abbreviations; cap, capitated processes of the laterosphenoid; f. bsf, facet for the basisphenoid; for?, foramen?; f. par, facet for the parietal; f. pro, facet for the protic; i. cav, internal cavity of the laterosphenoid.

The lateral surface does not seem to have any crest on its external surface, as in most basal crocodyliforms (Pol et al., 2014) and “sphenosuchians” where this region is preserved (i.e., Sphenosuchus, Dibothrosuchus, Junggarsuchus and Almadasuchus). The lateral surface of the laterosphenoid is convex in general, as it forms the anterolateral border of the lateral wall of the forebrain. This convexity is particularly developed anteriorly, where it is markedly beveled. On the posterior half of the lateral surface an elongated, foramen leading dorsally into the bone is present (Fig. 12C). However, this foramen does not communicate with the neurocranial cavity as it is closed by a medial lamina.

Phylogenetic relationships

In order to evaluate the affinities of Macelognathus the latter taxon was included in a phylogenetic dataset. The chosen dataset was the one used by Pol et al. (2013), as this matrix includes the highest amount of “sphenosuchian” taxa and also has an expanded crocodyliform sampling when it is compared to other datasets (e.g., Clark, Sues & Berman, 2000; Clark et al., 2004; Nesbitt, 2011). However, considering recent contributions on crocodylomorph phylogenies (Nesbitt, 2011; Zanno et al., 2015) we expanded this matrix including new characters and new taxa. As a result, our new dataset is composed of 39 taxa and 138 characters.

Taxon sampling

The new taxa incorporated are: the paracrocodylomorph Saurosuchus galilei Reig, 1959; and the crocodylomorphs CM 73372 (see Nesbitt, 2011), Carnufex carolinensis Zanno et al., 2015, Redondavenator quayensis Nesbitt, Irmis & Lucas, 2005, Trialestes romeri (Reig, 1963), Hallopus victor, Macelognathus vagans, and Hemiprotosuchus leali Bonaparte, 1969. Saurosuchus is added to increase the number of outgroups in the analysis, as Wilberg (2015) noted that the inclusion of more outgroups might affect the results. Also, Saurosuchus represents a very well-known paracrocodylomorph with both reported cranial (Alcober, 2000) and postcranial remains (Trotteyn, Desojo & Alcober, 2011).

Among the crocodylomorphs included, CM 73372 was recently found as the one of the basalmost members of the clade (Nesbitt, 2011), and thus is relevant to the following analysis. Other taxa relevant in this matter are two putative crocodylomorphs relatively recently published: Redondavenator quayensis (Nesbitt et al., 2005) and Carnufex carolinensis(Zanno et al., 2015; Drymala & Zanno, 2016). In Zanno et al.’s (2015) original analysis Carnufex was recovered at the base of Crocodylomorpha, in a polytomy with CM 73372, while Redondavenator was depicted as the sister taxon of Sphenosuchus. The sister group relationship between Sphenosuchus and Redondavenator was supported by a single synapomorphy: an elongated maxillary process on the premaxilla (Nesbitt, 2011; char. 2-1). It is important to consider that this character cannot be scored in Sphenosuchus, as Walker stated in his seminal paper that the dorsal end of this process is not preserved (Walker, 1990, p. 12; SAM-PK 3014). However, in a second analysis including both Carnufex and Redondavenator Drymala & Zanno (2016) recovered both taxa at the base of Crocodylomorpha together with CM 73372.

On the other hand, Trialestes and Hallopus have not been included in any analysis in almost the past 30 years. Only Trialestes was incorporated in the precursor dataset of Benton & Clark (1988). Finally, Macelognathus was added in Pol et al.’s (2013) dataset, but it was recovered as a wildcard taxon in that analysis and was removed from the consensus. However, Pol et al. (2013) mentioned that Macelognathus was recovered as the sister group of Almadasuchus, Terrestrisuchus + Litargosuchus or Saltoposuchus on the different most parsimonious trees obtained in their analysis. Those topologies were mostly supported by femoral characters, as the braincase of Macelognathus was not yet described. Hemiprotosuchus leali represents the only crocodyliform added in this analysis as it represents the oldest record of the clade (Late Triassic; Los Colorados Formation, La Rioja, Argentina; Bonaparte, 1972a).

Finally, it is relevant to discuss which materials were considered for some of the operational taxonomical units (OTUs) of our analysis. For a full list of the specimens examined the reader is referred to the Supplemental Information. The two most taxonomically controversial taxa are Trialestes romeri and Hesperosuchus agilis. Trialestes has been involved in a controversy since its original description by Reig (1963). Two specimens have been assigned to this taxon (PVL 2561, 3889), but other authors recognized that one specimen could represent a dinosaur (Clark, Sues & Berman, 2000) or even another crocodylomorph (Ezcurra, Lecuona & Irmis, 2008). In consequence, to avoid future problems with our codifications of the taxon, we restrict our scorings of Trialestes to the type specimen (PVL 2561), which also preserves clear crocodylomorph synapomoporphies (e.g., elongated proximal carpals). The other controversial taxon is Hesperosuchus agilis Colbert, 1952, especially regarding its content. Colbert (1952) established the taxon on a well preserved but rather fragmentary specimen from the lower part of the Chinle Formation. Later on, Clark, Sues & Berman (2000) described a much more complete individual from the upper part of the Chinle Formation and assigned it Hesperosuchus. However, Nesbitt (2011) and Irmis, Nesbitt & Sues (2013) thought this to be unlikely as it would extend the range of the same species through the entire deposition of the Chinle Formation, possibly as long as 20 million years. In later contributions other authors have mentioned and even scored some differences between both specimens (Drymala & Zanno, 2016). However, we see little differences between both specimens and many of these morphological differences are indeed present both in the type (AMNH FR 6756) and the referred specimen (CM 29894) (e.g., see characters 137 and 138). Furthermore, both specimens and another specimen assigned to H. “agilis” (UCMP 129470) share the development of sharp postzygodiapophyseal laminae, a trait not present in other North American “small” crocodylomorphs (i.e., not in Carnufex). As a result, we base our scorings of Hesperosuchus agilis on both the type (AMNH FR 6756) and the Carnegie Museum (CM 29894) specimens.

Character sampling

A total of 40 characters were added to Pol et al.’s (2013) dataset of 96 characters scored for 32 taxa. Among these 13 are new and the remaining 27 are from other recent datasets, where the majority of them come from Nesbitt’s (2011) contribution on archosaurian relationships. As new characters were formulated, and many of the characters from other contributions were modified (either in their formulation or scorings), an explanation of them is given below. Our focus on this discussion is on non-crocodyliform taxa, but for more information on the crocodyliform specimens used to evaluate these characters, the reader is referred to the Supplemental Information.

97. Supraoccipital: fused with the exoccipital (0); or, as a separate ossification (1) (NEW) (Figs. 13A, 13C and 13D). In basal pseudosuchians (Gracilisuchus, MCZ 4117; Stagonolepis, Walker, 1961) and paracrocodylomorphs (Saurosuchus, PVSJ 32; Postosuchus (Weinbaum, 2011)) the supraoccipital is fused with the paroccipital processes, not being present as a separate ossification. In crocodylomorphs where this region of the skull is known (e.g., Terrestrisuchus (Crush, 1984), Kayentasuchus, UCMP 131830, Almadasuchus, MPEF-PV 3838; P. richardsoni, UCMP 131827) the supraoccipital is a separate ossification.

Figure 13 Selected phylogenetic characters used in this contribution.

Skulls of (A) Kayentasuchus walkeri (UCMP 131830); (C) Dibothrosuchus elaphros (IVPP V 7907); and, (E) Almadasuchus figarii (MPEF PV 3838) in posterior views. Details of the supratemporal fenestra in dorsal view of (B) Litargosuchus leptorhynchus (BP/1/5237); (D) Hesperosuchus agilis (CM 29894); and, (F) Almadasuchus figarii (MPEF PV 3838). See Character Sampling and Supplemental Information for details.

98. Supraoccipital shape: narrow, being dorsoventrally taller than lateromedially wide (0); or, wide, being lateromedially wider than dorsoventrally high (1) (NEW) (Figs. 13A, 13C and 13D). In most crocodylomorphs the supraoccipital bone is rectangular in posterior view, as it is taller than wide, such as Pseudhesperosuchus (PVL 3830), Hesperosuchus (CM 29894), Terrestrisuchus (Crush, 1984), Kayentasuchus (UCMP 131830), Sphenosuchus (SAM-PK 3014), Dibothrosuchus (IVPP V 7907) and Junggarsuchus (IVPP V 14010). On the other hand, Almadasuchus (MPEF-PV 3838), Macelognathus (LACM 5572/150148, see description), crocodyliforms (e.g., Protosuchus haughtoni (BP/1/4726, 4746, 4770), Gobiosuchus Osmólska, Hua & Buffetaut, 1997) and most thalattosuchians (e.g., Teleosaurus (Jouve, 2009); Pelagosaurus (BSP 1890); Metriorhynchus superciliosus Blainville, 1853 (AMNH 997); Dakosaurus andiniensis Vignaud & Gasparini, 1996 (Pol & Gasparini, 2009); Cricosaurus araucanensis (MLP 72-IV-7-1)) have a wide supraoccipital bone, where this element is wider than tall.

99. Quadrate-Laterosphenoid contact: absent (0); or, present (1) (modified from Clark, 1986) (Figs. 13B and 13F). This character is a slight modification in its phrasing compared to how it was originally stated, as it only made reference to the quadrate head. We agree with Clark’s (1986) observations, as it was scored as present in crocodyliforms with the exception of thalattosuchians. Among sphenosuchians this condition is only present in Almadasuchus (MPEF-PV 3838). See also character 104.

100. Basioccipital recesses: absent (0); or, present as paired foramina located in a median deep depression on the ventral surface of the bone (1) (NEW). In paracrocodylomorphs (e.g., Postosuchus (Weinbaum, 2011); Saurosuchus, PVSJ 32) the basioccipital does not bear any ventral depression anterior to the occipital condyle. In most non-crocodyliform crocodylomorphs the basioccipital bears ventral a median depression, which is divided in two parallel tubes internally. These are observed in Hesperosuchus (CM 29894), Sphenosuchus (SAM-PK 3014), Dibothrosuchus (IVPP V 7907) Macelognathus (LACM 5572/150148) and Almadasuchus (MPEF-PV 3838). Crush (1984, p. 138) describes a well-developed basioccipital recess for Terrestrisuchus, although he mentions it is square shaped. The basioccipital is not recessed in Junggarsuchus (IVPP V 14010) and crocodyliforms (e.g., P. richardsoni, UCMP 131827), although in the latter forms the posterior margin of the median Eustachian tubes.

101. Length of the posterodorsal process of the postorbital: short, not reaching the midlength of the supratemporal fenestra (0); or, long, exceeding the midlength of the supratemporal fenestra (1) (NEW) (Figs. 13B, 13D and 13F). As in most basal pseudosuchians (Postosuchus (Weinbaum, 2011); Saurosuchus, PVSJ 32), crocodyliforms generally have a short posterodorsal process of the postorbital, very short and not exceeding the midlength of the supratemopral fenestra (e.g., P. richardsoni, MCZ 6727; P. haughtoni, BP/1/4726;Orthosuchus, SAM-PK 409, C. yacare). This condition is also present in the “sphenosuchians” Terrestrisuchus (Crush, 1984) and Dibothrosuchus (IVPP V 7907). In Pseudhesperosuchus (PVL 3830), Hesperosuchus (CM 29894), Sphenosuchus (SAM-PK 30141), Junggarsuchus (IVPP V 14010) and Almadasuchus(MPEF-PV 3838) the posterodorsal process of the postorbital is posteriorly elongated, reaching more than half the anteroposterior development of the supratemporal fenestra.

102. Quadrate fenestra: with participation of the quadratojugal (0); or, exclusively bounded by the quadrate (1) (NEW). Pneumatic quadrate fenestrae have been recognized in the crocodylomorphs Terrestrisuchus, Junggarsuchus, Almadasuchus, Macelognathus and Dibothrosuchus (see Description and char. 46 in the Supplemental Information). However, besides size variation, there is variation in the contribution of the surrounding bones in the different taxa. Terrestrisuchus(Crush, 1984) has at least some participation of the quadratojugal on the anterior end; while in Junggarsuchus (IVPP V 14010), Almadasuchus (MPEF-PV 3838) and crocodyliforms (e.g., Protosuchus richardsoni, UCMP 131827; Gobiosuchus (Omolska, 1997)) the quadrate fenestrae are completely surrounded by the quadrate. As previously mentioned, this condition cannot be precisely known in Dibothrosuchus (IVPP V 7907) and Macelognathus (LACM 5572/150148) due to the incomplete preservation of the quadrate fenestra.

103. Number of quadrate fenestrae: one (0), or more than one (1) (Clark, 1994). Crocodylomorphs that have quadrate fenestrae usually have one on the main body of the quadrate. However, Junggarsuchus is unusual in this aspect, as at least two fenestrae are identified on its quadrate (Klein et al., unpublished data). This condition is similar to the one present in basal crocodyliforms that have multiple quadrate fenestrae (e.g., Protosuchus richardsoni (UCMP 131827)).

104. Prootic: exposed in dorsal view, on the supratemporal fossa (0); or not exposed in dorsal view (1) (NEW) (Figs. 13B, 13D and 13F). In basal pseudosuchians (Postosuchus (Weinbaum, 2011); Saurosuchus, PVSJ 32; Gracilisuchus, MCZ 4117) and non-crocodyliform crocodylomorphs (e.g., Hesperosuchus, CM 29894; Litargosuchus, BP/1/5237; Dibothrosuchus, IVPP V 7907; Almadasuchus, MPEF-PV 3838) the prootic is exposed on the posterodorsal region of the supratemporal fossa. Most crocodyliforms lack the prootic exposed dorsally on the supratemporal fossa (e.g., P. richardsoni, UCMP 131827; Orthosuchus, SAM-PK 409) due to the broad contact of the quadrate with the laterosphenoid. However, Almadasuchus has a dorsally exposed prootic on the posterodorsal region of the supratemporal fossa, but has a quadrate-laterosphenoid contact (see char. 99). On the other hand, thalattosuchians (e.g., Cricosaurus araucanensis, MLP 72-IV-7-1; Teleosaurus (Jouve, 2009); Pelagosaurus (Pierce & Benton, 2006)) exhibit the same condition as in non-crocodyliform crocodylomorphs.

105. Exit of cranial nerves IX–XI: exit the braincase ventromedially (0); or, through a common foramen on the ventromedial region of the paroccipital process (vagus foramen) (1) (NEW) (Fig. 13E). The primitive condition for Crocodylomorpha is the exit of the cranial nerves IX–XI through the ventrolateral region of the skull, thus not leaving any distinct foramen (e.g., Gracilisuchus, MCZ 4117; Postosuchus, (Weinbaum, 2011); Saurosuchus, PVSJ 32). This condition is present in most “sphenosuchian” crocodylmorphs, like Pseudhesperosuchus (PVL 3810), Hesperosuchus (CM 29894; AMNH FR 6758), Terrestrisuchus (Crush, 1984), Litargosuchus (BP/1/5237), Kayentasuchus (UCMP 131830), Sphenosuchus (SAM-PK 3014), Dibothrosuchus (IVPP V 7907) and Junggarsuchus (IVPP V 14010). Almadasuchus (MPEF-PV 3838) and crocodyliforms (e.g., P. richardsoni, UCMP 131827; Protosuchus haughtoni (BP/1/4726, 4746, 4770), Orthosuchus, SAM-PK 409) exhibit a foramen on the ventrolateral region of the paroccipital process that has been interpreted widely as the exit of the nerves IX–XI (i.e., vagus foramen). This condition is also present in thalattosuchians (e.g., Pelagosaurus (Clark, 1986; Pierce & Benton, 2006); Teleosaurus (Jouve, 2009); Dakosaurus (Pol & Gasparini, 2009)).

106. Postzygodiapophyseal laminae on the posterior cervical and anterior dorsal vertebrae: absent or low (0); or, present as sharp lamina delimiting a pit posterior to them on the neural arch (1) (NEW) (Figs. 14B and 14C). Sharp postzygodiapophyseal laminae are present on the posterior cervical and anterior dorsal vertebrae of paracrocodylomorph archosaurs (Saurosuchus, PVL 2198; Fasolasuchus, PVL 3950; Postosuchus (Long & Murry, 1995)), Carnufex (Drymala & Zanno, 2016) and Hesperosuchus (AMNHFR 6576, CM 29894, UCMP 129470). Posterior to these laminae, a well-marked semicircular depression is limited. Dibothrosuchus (IVPP V 7907) has a similar condition on its anterior dorsal vertebrae, although the postzygodiapophyseal lamina is not as sharply projected as in Hesperosuchus.

Figure 14 Selected phylogenetic characters related to Hesperosuchus agilis used in this contribution.

(A) Maxilla of the type specimen of Hesperosuchus agilis (AMNH FR 6758) in lateral view. Posterior cervical vertebrae in lateral view of: (B) the type (AMNH FR 6758); and, (C) the referred (CM 29894) specimen of Hesperosuchus agilis. Proximal radii in anterior view of: (D) Pseudhesperosuchus jachaleri (PVL 3830); (E) the type specimen of Hesperosuchus agilis (AMNH FR 6758); and, the referred (CM 29894) specimens of Hesperosuchus agilis. See characters 106, 137, and 138.

107. Length of the radius: shorter than the humerus (0); or, longer than the humerus (1) (NEW). This trait was originally highlighted by Clark, Sues & Berman (2000) referring to the particular elongation of the radius present in Trialestes. However, such elongation of the radius is more widely distributed than previously reported. Besides Trialestes (PVL 2561), longer radii than their respective humeri are also present in Litargosuchus (BP/1/5237), Dibothrosuchus (IVPP V 7907), Dromicosuchus (NCSM 13733) and variably in Hesperosuchus (present in AMNH FR 6758, while in CM 29894 the humerus is longer). On the other hand, most non-crocodylomorph pseudosuchians (e.g., Gracilisuchus, MCZ 4117; Postosuchus, (Weinbaum, 2011); Saurosuchus, PVSJ 32), Saltoposuchus (SMNS 12597), Junggarsuchus (IVPP V 14010), Hallopus (YPM 1914) and crocodyliforms (e.g., P. richardsoni, UCMP 131827; Protosuchus haughtoni (BP/1/4726, 4746, 4770), Orthosuchus, SAM-PK 409) have longer humeri than their radii. The relative length of the anterior zeugopodium seems not to be correlated with the lenghts of the posterior one. Taxa such as Dromicosuchus (Sues et al., 2003) and Hallopus (Walker, 1970) have femora proximodistally shorter than the tibiae (see char. 32, Supplemental Information), while their radii are longer than the humeri.

108. Proximomedial process of the radiale: absent (0); or, present (1) (NEW). Besides the elongation present in this element (Clark, 1986; Benton & Clark, 1988), crocodylomorphs are also characterized by the presence of a proximomedial process on the radiale. This process articulates with the ulna and the ulnare. As with the elongation on the proximal carpals, this process is also lacking in CM 73372. The presence of the proximomedial process of the radiale might be thought to be correlated with the elongation of the proximal carpals, as these traits have the same distribution among crocodylomophs. However, this can be very useful in taxa where the radiale is not preserved completely, and yet represents a crocodylomorph synapomorphy with the exclusion of the CM 73372.

109. Anterior process of the squamosal: elongated, less than one third of the lateromedial width of the supratemporal fossa (0); transversely broad, more than one third the width of the supratemporal fossa (1); or, very broad, as wide as the width of the supratemporal fossa (modified from Clark, 1986; Sereno & Wild, 1992; Nesbitt, 2011) (Figs. 13B, 13D and 13F). Clark (1986) recognized the lateromedially expanded anterior process of the squamosal as a crocodylomorph synapomorphy. Sereno & Wild (1992) followed the same idea, and established that unlike other pseudosuchians, in crocodylomorphs the anterior process of the squamosal was at least one third of the posterior width of the whole skull. Finally, Nesbitt (2011) observed a similar, but more restricted variation among some crocodylomorphs (Litargosuchus, P. richardoni, Orthosuchus and Alligator), where in these taxa the anterior process of the squamosal was as transversely wide as the supratemporal fenestra. We integrated these independent characters in a single, multistate character, and using as a comparative value the width of the supratemporal fenestra. As previously noted, crocodylomorphs have a lateromedially expanded anterior process of the squamosal. This condition is present in several “sphenosuchians” (Pseudhesperosuchus, PVL 3810;Hesperosuchus, CM 29894; Sphenosuchus, SAM-PK 3014; Dibothrosuchus, IVPP V 7907; Junggarsuchus, IVPP V 14010; Almadasuchus, MPEF-PV 3838) and most crocodyliforms (Hsisosuchus (Li, Wu & Li, 1994); Sichuanosuchus (Wu, Sues & Dong, 1997); Simosuchus, UA 8679). Finally Litargosuchus (BP/1/5237), Terrestrisuchus (Crush, 1984), Protosuchus (P. richardsoni, UCMP 131827; Protosuchus haughtoni (BP/1/4726, 4746, 4770)), Notosuchus (MACN-RN 1037) and Alligator (Iordansky, 1973) have a very wide anterior process of the squamosal, as this process is as wide as the supratemporal fossa. Kayentasuchus(UCMP 131830) was scored as having an expanded anterior process of the squamosal, but we could not determine to what extent (1/2), as the preservation of the specimen does not allow it. Finally, we scored the condition of Orthosuchus as missing data, as the anterior process of the squamosal is not completely preserved (SAM-PK 409).

110. Lateral extent of the paroccipital processes: ends lateral to the lateral border of the supratemporal fenestra (including the fossa) (0); or, ends medial to or at the margin of the border of the supratemporal fenestra (1) (modified from Nesbitt, 2011) (Figs. 13A, 13C and 13E). In Nesbitt’s study (2011) the derived condition was found in a group of derived crocodylomorphs, which included Kayentasuchus, Litargosuchus, Protosuchus and Orthosuchus. However, due to difficulties in scoring this character we also included the lateral limit of the supratemporal fossa, which was not stated in the original character. Without taking into account the supratemporal fossa most crocodylomorphs would be scored as having the lateral end of the paroccipital processes lateral to the border of the supratemporal fenestra/fossa, even some of them scored otherwise by Nesbitt (2011). However, when the distribution of this character was revised several differences were found. Paroccipital processes with short lateral projection (i.e., not reaching the lateral border of the supratemporal fossa or fenestra) are present in Pseudhesperosuchus (PVL 3810), Litargosuchus (BP/1/5237) and Hesperosuchus (CM 29894), among non-crocodyliforms crocodylomorphs. Kayentasuchus (UCMP 131830) bears laterally projected paroccipital processes, as these almost reach the lateralmost border of the skull in occipital view (Clark & Sues, 2002). Paroccipital processes that end more laterally than the lateral edge of the supratemporal fossa/fenestra are also present in Protosuchus (P. richardsoni (UCMP 131827); Protosuchus haughtoni (BP/1/4726, 4746, 4770)), as in this genus the supratemporal fenestrae lack a fossa on their lateral part, thus having little lateral development. Orthosuchus also has this state of the character, but unlike Protosuchus the paroccipital processes have a great lateral projection reaching the lateral end of the skull in occipital view (SAM-PK 409; Nash, 1975). Among crocodyliforms there is not a clear trend as laterally short paroccipital processes are present in Zosuchus (Pol & Norell, 2004), some notosuchians (Baurusuchus (Carvalho, Campos & Nobre, 2005); Simosuchus (Kley et al., 2010)) and thalattosuchians.

111. Anterior end of the dentaries: tapering to a point (0); or, dorsally expanded, forming a distinct step (1) (modified from Nesbitt, 2011). This trait was used to refer CM 29894 to Hesperosuchus by Clark, Sues & Berman (2000). However this observation was questioned later as an anteriorly expanded dentary was also present in Postosuchus and Dromicosuchus (Nesbitt, 2011). We agree with the presence of such an expansion in Postosuchus (Weinbaum, 2011), but on the other hand, this condition cannot be observed in Dromicosuchus (NCSM 13733) as the mandibles are closed and articulated in the only known specimen. Trialestes (PVL 2561) also has an anteriorly expanded dentary, but this condition should be handled with care considering the aggressive mechanical preparation that specimen suffered.

112. Anterior part of the dentary: bears teeth (0); or, edentulous (1) (Parrish, 1994). Most archosauriforms have teeth along their entire tooth row. Exceptions among pseudosuchians are some poposauroids (Effigia (Nesbitt, 2007); Lotosaurus (Nesbitt, 2011)), aetosaurs (Stagonolepis (Walker, 1961)), and among crocodylomorphs Macelognathus (YPM 1415; LACM 5572/150148).

113. Acromial process of scapula: in the same plane of the proximal surface of the scapula (0); or, distinctly raised, forming an abrupt step between the scapular blade and the proximal end of the scapula (1) (modified from Nesbitt, 2011). Contrasting with the condition present in basal archosauriforms, archosaurs have an expanded acromial process of the scapula, where this region forms a distinct anterior step (Leardi, 2013). This condition is reversed in paracrocodylomorph pseudosuchians (e.g., Effigia (AMNH 30587); Postosuchus (Long & Murry, 1995)) and Hesperosuchus (AMNH 6758). Other crocodylomorphs where the proximal end of the scapula is adequately known bear expanded acromial processes (e.g., Trialestes, PVL 2561; Pseudhesperosuchus, PVL 3810; Dibothrosuchus, IVPP V 7907; Orthosuchus, SAM-PK 409).

114. Coracoid posteroventral edge: smooth (0); or, with a groove (1) (Nesbitt, 2011). This character evaluates the presence of a ventral groove on the distal end of the coracoids, which serves for the articulation of the interclavicle. This groove was noted in Postosuchus (missing in other rauisuchids) and non-crocodyliform crocodylomorphs with the exception of Litargosuchus (Nesbitt, 2011), with increased taxon sampling this groove is also observed to be absent in Junggarsuchus (IVPP V 14010) and Dibothrosuchus (IVPP V 7907).

115. Scapular contribution to the glenoid: lesser than the coracoid contribution (0); or, equal or greater than the coracoid contribution (1) (NEW). In most archosauriforms the glenoid is formed mostly by the coracoids, as can be seen in Riojasuchus (PVL 3827), Stagonolepis (Walker, 1961), Prestosuchus (UFGRS PV 0629T), and Postosuchus (Weinbaum, 2002). In crocodylomorphs, unlike other pseudosuchians, the glenoid is mostly formed by the scapula. This condition can be observed in Pseudhesperosuchus (PVL 3810), Hesperosuchus (AMNH FR 6758), Terrestrisuchus (Crush, 1984), Sphenosuchus (SAM-PK 3014), Dibothrosuchus (IVPP V 7907), Junggarsuchus (IVPP V 14010), Protosuchus (P. richardsoni, UCMP 131827; P. haughtoni (BP/1/4726, 4746, 4770)), among others.

116. Humeral proximal head: confined to the proximal surface (0); or, posteriorly expanded and hooked (1) (Nesbitt, 2011). Nesbitt (2011) formulated this character observing the similarities among the posterior projection of the humeral heads in rauisuchids (e.g., Postosuchus) and crocodylomorphs. A hooked posteriorly projecting humeral head is also present in basal crocodyliforms like Protosuchus and Orthosuchus, but not in more derived forms (Alligator in that analyses). We agree with those observations, and also did not observe that condition in other crocodyliforms.

117. Olecranon process on the ulna: present (0); or, absent or very low (1) (modified from Wu, Sues & Dong, 1997). The olecranon process is present in the proximal end of the ulna in most archosaurs (Leardi, 2013). This condition is maintained in most pseudosuchians (e.g., Stagonolepis (Walker, 1961); Postosuchus (Weinbaum, 2002)) and “sphenosuchians” (Hesperosuchus, AMNH 6758, CM 29894; Terrestrisuchus (Crush, 1984); Dromicosuchus (Sues et al., 2003); Sphenosuchus, SAM-PK 3014; Dibothrosuchus, IVPP V 7907; Hallopus, YPM 1914). In crocodyliforms (e.g., P. richardsoni, AMNH FR 3024; Orthosuchus, SAM-PK 409; Simosuchus, UA 8679; Alligator, MPEF AC 205) the olecranon process is absent or reduced, as in these taxa there is no marked discontinuity between the proximal surface and the posterior end of the proximal surface. A reduced olecranon process is also present in some non-crocodyliform crocodylomorphs: Trialestes (PVL 2561), Pseudhesperosuchus (PVL 3810), Saltoposuchus (SMNS 12597) and Junggarsuchus (IVPP V 14010).

118. Proximolateral process of the ulna: located at the midpoint of the proximolateral surface of the ulna (0); or, anteriorly displaced, at the level of the anterior process of the ulna (1) (NEW). The archosauriform ulna in proximal view has, generally, three proximal processes: an anteromedial, a lateral and a posterior process (usually the olecranon, when it is proximally projected). This condition is maintained in crocodylomorph outgroups (Stagonolepis (Walker, 1961); Postosuchus (Weinbaum, 2002)). In crocodylomorphs the lateral process of the ulna is displaced anteriorly, almost at the same level of the anterior process, leaving the articular surface for the radius exposed anteriorly. Pseudhesperosuchus (PVL 3810), Hesperosuchus (AMNH FR 6758), Sphenosuchus (SAM-PK 3014) and Caiman (MPEF AC 205) are examples of this condition among crocodylomorphs. In many taxa where the proximal surface of the ulna cannot be observed due to preservational features, this trait can also be identified as the anteroproximal end of the ulna is expanded.

119. Distal end of the ulna: anteroposteriorly compressed or rounded (0); or, with anterior expansion (1) (Nesbitt, 2011). Nesbitt (2011) discussed that, unlike other archosaurs, rauisuchids (e.g., Postosuchus, Batrachotomus) have an anteriorly expanded distal end of the ulna. We agree with that statement and the taxa in our analyses are coded that way.

120. Metacarpals II–V configuration: spreading (0); or, compact (1) (modified from Clark, Sues & Berman, 2000). We modified this character just focusing on the general disposition of the metacarpals, eliminating the elongation part of the description. This character was originally used by Sereno & Wild (1992) and later modified by Clark, Sues & Berman (2000) to try to reflect the digitigrade configuration of the manus of several “sphenosuchians”. As also discussed in the latter paper, the digitigrady condition relies on an interpretative reconstruction of the manus, and there is notable variation in the general configuration of the metacarpals of “sphenosuchians”, especially when compared with crocodyliforms. Also this character seeks to replace the one referring to the proximal configuration of the metacarpals (char. 30), which has proven to be controversial regarding its scorings, and recently no variation among crocodylomorphs has been recognized (Nesbitt, 2011), thus making this character non-informative. The main problem with how the character was restated is that it requires fairly complete and articulated metacarpals to properly allow differentiating among the different character states. In general, most pseudosuchians have a spreading configuration of their metacarpus, in which the metacarpals diverge gradually from their proximally imbricated proximal end towards their distal ends. This condition is present in aetosaurs (Stagonolepis (Walker, 1961); Longosuchus (TMM 31185-84a)) and Postosuchus (Peyer et al., 2008). The condition of CM 73372 is dubious despite that it was scored in recent analyses as having overlapping metacarpals (Nesbitt, 2011), as the metacarpus was disarticulated during mechanical preparation. In non-crocodyliform crocodylomorphs where a manus was found articulated a compressed metacarpus can be observed (Hesperosuchus, CM 29894; Saltoposuchus, SMNS 12352, 12597; Terrestrisuchus, NHMUK R7557 (Crush, 1984); Dibothrosuchus, IVPP V 7907; Junggarsuchus, IVPP V 14010; and Hallopus, YPM 1914). In contrast, crocodyliforms have a spreading metacarpus (e.g., P. richardsoni, MCZ 6727; Orthosuchus, SAM-PK 409; Sichuanosuchus (Wu, Sues & Dong, 1997); Notosuchus, MACN-RN 1037; Caiman, MPEF AC 205).

121. Dorsoventrally oriented crest dorsal to the supraacetabular crest: absent (0), or, present (1) (modified from Nesbitt, 2011). This character is a modification of a set of three characters postulated by Nesbitt (2011), which evaluated the presence, direction and thickness of a crest situated dorsally to the acetabulum (chars. 265-267). However, when taxon sampling is reduced to crocodylomorphs and outgroups, the morphological variation is reduced to a single morphology: a dorsoventrally oriented crest dorsal to the acetabulum. This crest is absent in basal pseudosuchians (Stagonolepis (Walker, 1961; Nesbitt, 2011); Gracilisuchus, PVL 4597 (Lecuona & Desojo, 2012)), but is present in paracrocodylomorphs (Saurosuchus (Nesbitt, 2011); Postosuchus (Weinbaum, 2002)) and basal crocodylomorphs (CM 73372; Hesperosuchus, AMNH 6758; Terrestrisuchus (Nesbitt, 2011); Dromicosuchus, NCSM 13733). This crest is absent Dibothrosuchus (IVPP V 7907), Macelognathus (LACM 5572/150148) and crocodyliforms (e.g., P. richardsoni (AMNH 3024), Orthosuchus (SAM-PK 409)).

122. Preacetabular process of the ilium: short and does not extend anteriorly to the acetabulum (0); or, elongated, but shorter than the postacetabular process (1) (modified from Galton, 1976; Clark, 1986; among others). An elongated preacetabular process (usually exceeding the anteriorly to the anterior border of the acetabulum) has been usually recognized as a crocodylomorph synapomorphy (Clark, 1986; Benton & Clark, 1988; Nesbitt, 2011). This trait is also present in basal crocodyliforms like P. richardsoni (AMNH FR 3024) and Orthosuchus (SAM-PK 409), but not in mesoeucrocodylians (e.g., Baurusuchus (Nascimento & Zaher, 2010); Alligator (Mook, 1921)) nor in the thalattosuchians sampled in our analysis (Andrews, 1913).

123. Ilium orientation: mainly vertical orientation (0°–20°) (0); or, ventrolaterally deflected about 45°(1) (Benton & Clark, 1988). Most archosauriforms have vertically oriented ilia. However, aetosaurs (e.g., Stagonolepis (Walker, 1961)) and paracrocodylomorphs (Saurosuchus, PVSJ unnumbered; Postosuchus (Weinbaum, 2002)) have been recognized to have ventrolaterally deflected ilia (Benton & Clark, 1988). Nesbitt (2011) recognized this trait also in CM 73372, Dromicosuchus (NCSM 13733) and Kayentasuchus (UCMP 131830). Nesbitt scored a ventrolaterally oriented ilium in Hesperosuchus(AMNH FR 6758), but that specimens has a very badly preserved ilium and was scored as missing (?) in this analysis.

124. Ventral margin of the acetabulum: convex (0); or, concave (1) (modified from Nesbitt, 2011). Nesbitt (2011) erected this character as a modification of the classic character treating the perforate condition of the acetabulum (Bakker & Galton, 1974). Convex ventral margin of the acetabulum is present in basal pseudosuchians (Gracilisuchus, PVL 4597 (Lecuona & Desojo, 2012); Stagonolepis (Walker, 1961)), Saurosuchus (Nesbitt, 2011), Postosuchus (Weinbaum, 2002), Hesperosuchus (Nesbitt, 2011) and Dromicosuchus (NCSM 13733). Other crocodylomorphs (e.g., Terrestrisuchus (Crush, 1984); Dibothrosuchus, IVPP V 7907; Macelognathus, LACM 5572/150148) and crocodyliforms (e.g., P. richardsoni, AMNH 3024; Orthosuchus, SAM-PK 409) have concave ventral margins of the acetabulum.

125. Ilium dorsal margin dorsal to the supraacetabular rim: rounded or sharp (0); or, flat (1) (Nesbitt, 2011). Nesbitt proposed that the dorsal margin of the iliac blade in crocodylomorphs is flat when it is compared to more basal pseudosuchians (e.g., Saurosuchus, Postosuchus) and CM 73372. The only mesoeucrocodylian scored in that analysis has a rounded dorsal margin of the iliac blade (Alligator). We agree with those scorings and incorporate Macelognathus, which has the “sphenosuchian condition”. Furthermore, the dorsal margin of the ilia of Notosuchus (MACN-RN 1044) and Baurusuchus (UFRJ DG 285-R) are rounded.

126. Obturator foramen on the pubis: present (0); or, absent (1) (modified form Sereno & Wild, 1992). In previous contributions, the size of the obturator foramen has been recognized to vary among crocodylomorphs and closely related groups (Sereno & Wild, 1992; Nesbitt, 2011). The main focus on this structure was the relative size of the obturator foramen, as it was noted that it was “enlarged” when it was compared to the condition present in rauisuchians and other pseudosuchians. However, this rather ambiguous state of character (the enlarged condition) raised some doubts about the scoring process and, also, is problematic when in a multistate character with absence of a foramen (Brazeau, 2011). In this analysis we chose a simpler solution, which is to simply score the presence of the obturator foramen. An obturator foramen is present in basal pseudosuchians (Gracilisuchus, PVL 4597 (Lecuona & Desojo, 2012); Stagonolepis (Walker, 1961)), and Saurosuchus (Nesbitt, 2011). Among crocodylomorphs this foramen is also present in Hesperosuchus (YPM 41198), Saltoposuchus (SMNS 12596) and Terrestrisuchus (Crush, 1984). The pubis of Dromicosuchus (NCSM 13733) seems to have preserved the anterior margin of a foramen (the obturator foramen) on the proximal end of the pubis. However, to avoid speculation Dromicosuchus was still coded as unknown (?). Unfortunately, the data on the crocodylomorphs most closely related to crocodyliforms is lacking, as in these forms (Sphenosuchus, Dibothrosuchus, Junggarsuchus, Almadasuchus, Macelognathus, Hallopus) no remains of the proximal end of the pubis have been reported. In contrast, all known crocodyliforms lack an obturator foramen on their pubes (e.g., P. richardsoni, AMNH 3024; Orthosuchus, SAM-PK 409; Hsisosuchus, Li, Wu & Li, 1994; Simosuchus, FMNH PR 2596; Alligator (Mook, 1921)). Thalattosuchians also lack an obturator foramen (Andrews, 1913).

127. Ischium medial contact with its antimere: all along its medial margin, but excluding the proximal end (0); or, restricted to the medial edge of the distal part (1) (modified from Nesbitt, 2011). This character was modified for use in the current study, as it had an extra state which was not informative in our analysis. Also, for the sake of clarity, state 1 (0 in Nesbitt, 2011) was slightly restated. Paracrocodylomorphs (e.g., Saurosuchus (Nesbitt, 2011); Postosuchus (Weinbaum, 2002)) have a very wide contact between the ischia, as it extends all along their medial margin only excluding the proximal ends. The same condition is observed in CM 73372 and in Terrestrisuchus (Crush, 1984). In Litargosuchus (BP/1/5237), thalattosuchians (Andrews, 1913; Pierce & Benton, 2006) and crocodyliforms (e.g., P. richardsoni, AMNH FR 3024; Orthosuchus, SAM-PK 409; Alligator (Mook, 1921)) the contact between the ischia is distally restricted, as it leaves a proximal portion (besides the proximal end) that is not involved in such contact. Unfortunately, the information available for non-crocodyliform crocodylomorphs is only known in the above mentioned taxa (CM 73372, Terrestrisuchus and Litargosuchus).

128. Ischium, distal end: plate-like (0); or, rounded (1) (Nesbitt, 2011). Nesbitt (2011) noted that many paracrodylomorphs (Saurosuchus and Postosuchus in this study) and CM 73372 had rounded distal ends of the ischia, unlike the condition in crocodylomorphs where the distal end of the ischium is lateromedially compressed. His codings are followed here.

129. Femur, proximal condylar fold: absent (0); or, present (1) (Nesbitt, 2011). Originally defined by Brochu (1992) and later incorporated as a character by Nesbitt (2011), the proximal fold has been recognized in paracrocodylomorphs (including crocodylomorphs) but not in pseudosuchians outside this clade.

130. Fibula, proximal end: rounded or elliptical (0); or, mediolaterally compressed (1) (Nesbitt, 2011). Nesbitt (2011) recognized that a mediolaterally compressed proximal end of the fibula was a shared derived trait of Postosuchus + crocodylomorphs. We agree with those scorings and the observations from an expanded taxon sampling of crocodyliforms agrees with that observation.

131. Pedal digit IV, number of phalanges: five (0); or, four (1) (Parrish, 1993). Parrish (1993) recognized the presence of four phalanges in pedal digit four as a crocodylomorph synapomorphy. However, data on this character is very scarce among non-crocodyliform crocodylomorphs as it is only known in CM 73372 and Terrestrisuchus. CM 73372 has five phalanges on pedal digit five as in Stagonolepis (Walker, 1961), Postosuchus (Peyer et al., 2008). Terrestrisuchus is the only “sphenosuchian” where this condition is known, and it has four phalanges as in P. richardsoni and other crocodyliforms (Alligator (Mook, 1921)).

132. Pedal digit V, number of phalanges: one or more (0); or, none (1) (modified from Gauthier, 1986; Parrish, 1993). This character was modified as it is usually considered in large datasets where ornithodirans are also included (Gauthier, 1986; Parrish, 1993; Nesbitt, 2011; among others). Although the number of phalanges on digit V is rather difficult to evaluate, as these can be easily disarticulated, this character can also be evaluated when metatarsal V is well-preserved. In taxa that lack phalanges on digit V, metatarsal V tapers to a point and does not have a distal articular surface for the proximal phalanx. Presence of at least one phalanx in pedal digit V is widespread among pseudosuchians, such as Stagonolepis (Walker, 1961), Gracilisuchus (PVL 4597 (Lecuona & Desojo, 2012)), Saurosuchus (PVL 2557) and Postosuchus (Peyer et al., 2008). At least one phalanx is also present in pedal digit V in CM 73372, Terrestrisuchus (Crush, 1984) and Hesperosuchus (YPM 41198, identified by the presence of a metatarsal V with a distal articular facet). On the other hand, Litargosuchus (BP/1/5237), Hallopus (YPM 1914), P. richardsoni (AMNH FR 3024), Orthosuchus (SAM-PK 409) and other crocodyliforms (e.g., Simosuchus, Baurusuchus, Alligator) have no phalanges on pedal digit V.

133. Ventrally open notch on ventral edge of rostrum at premaxilla-maxilla contact: absent (0), present as a notch (1), or present as a large notch (2), or present as a notch that is closed ventrally (or largely constrained at its ventral edge) (3) (modified from Clark, 1994). The presence of a notch to accommodate an enlarged mandibular tooth (or even two in some taxa) at the premaxilla-maxilla contact is a highly variable trait among crocodylomorphs, which is reflected in the large number of states for this character. Taxa closely related to crocodylomorphs lack a notch on the ventral edge of the premaxilla-maxilla suture (e.g., Saurosuchus, PVL 32; Postosuchus (Weinbaum, 2011)). On the other hand, Hesperosuchus (CM 29894, YPM 41198), Kayentasuchus (UCMP 131830), Dibothrosuchus (IVPP V 7907), P. richardsoni (MCZ 6727) and Orthosuchus (SAM-PK 409) have large notches that are partially closed ventrally. In other non-crocodyliform crocodylomorphs this region is partially damaged, and this character cannot be scored properly. Junggarsuchus (IVPP V 14010) is unique in lacking the notch entirely.

134. Nasal-frontal contact: transverse (0); or, frontals taper to a point (1) (Nesbitt, 2011). Nesbitt (2011) noticed that as in crocodylomorphs, in rauisuchians (represented by Postosuchus in our dataset) the suture between the nasals and frontals was not lateromedially straight, as the frontals wedged between the nasals and tapered to a point. In a few crocodyliforms this trait is reversed to the primitive condition (Gobiosuchus (Osmólska, Hua & Buffetaut, 1997); Simosuchus, UA 8679).

135. Ectopterygoid: single head (0); or, double headed (1) (Weinbaum & Hungerbühler, 2007). The ectopterygoid of Postosuchus and some basal crocodylomorphs was noted to have a subdivided lateral head for the articulation with the maxilla and the jugal (Weinbaum & Hungerbühler, 2007; Nesbitt, 2011).

136. Ectepicondylar groove: present (0); or, absent (1) (Benton & Clark, 1988). This character has been used in several archosaur studies (e.g., Benton & Clark, 1988; Nesbitt, 2011). The absence of an ectepicondular groove has been previously recognized in crocodylomorphs (e.g., Trialestes, PVL 2561; Hesperosuchus, AMNH FR 6758, CM 28984; Sphenosuchus, SAM PK 3014; Junggarsuchus, IVPP V 14010; P. richardsoni AMNH FR 3024). With the increased taxonomical sample this distribution changes as an ectepicondilar groove is present in the referred specimen (NCSM 21623) the “large bodied” crocodylomorph Carnufex (Drymala & Zanno, 2016).

137. Antorbital fossa: (0) well defined anterodorsally, yet not well defined along entire length of posterior process of maxilla; (1) well defined, forming complete circumference within the antorbital fenestra (modified from Drymala & Zanno, 2016). This character was modified removing a state (poorly defined), as it can not be scored in any of the taxa included in our analysis. Drymala & Zanno (2016) recognized the presence of a well defined antorbital fossa, all around its circumference, in what they defined as the “group H” which clustered the referred specimens of Hesperosuchus (H. “agilis” sensu Nesbitt, 2011 and Irmis, Nesbitt & Sues, 2013) and Dromicosuchus. Among basal crocodylomorphs, this trait was also observed in Kayentasuchus (Drymala & Zanno, 2016). Additionally, the holotype specimen of Hesperosuchus agilis (AMNH FR 6758) was scored as having an antorbital fossa not well delimited on its posteroventral edge, and used to differentiate the type specimen from the referred ones as the they formed a paraphyletic group (Drymala & Zanno, 2016). However, this trait can not be observed in the type specimen of H. agilis (AMNH FR 6758) as the posterior most region of the facial lamina of the maxilla is not preserved. In addition, the posterior part of the maxilla of the type specimen bears a well defined crest that can be interpreted as the border of the antorbital fossa (Fig. 14A). Thus, the differences noted on the antorbital fossa of the type and referred specimens of H. agilis are disregarded here.

138. Radius, proximal end, medial process: (0) absent, giving the radius a symmetrical aspect in anterior view; (1) present, giving the radius an asymmetrical aspect in anterior view (modified from Drymala & Zanno, 2016) (Figs. 14D–14E). Drymala & Zanno (2016) erected this character based on the morphology of the referred specimen of H. agilis (CM 29894) and an as yet unpublished taxon. We modified the character in order to clarify the different states, as a distinct medial process of the proximal end of the radius was ambiguous. In most crocodylomorphs, the proximal end of the radius is almost symmetrical in anterior view (Fig. 14D), while in CM 29894 the medial end is more expanded that the lateral one (Fig. 14F). Unlike Drymala & Zanno’s (2016) observations, the holotype specimen of H. agilis (AMNH FR 6758) also bears a well-projected proximomedial process of the radius, giving the proximal end of the radius an asymmetrical aspect in anterior view (Fig. 14E). As a result, all specimens of Hesperosuchus share this particular proximal radial morphology.

Although it was not included in our analysis, it is worth discussing a character erected by Nesbitt (2011). Nesbitt’s (2011) analysis and later datasets that use his matrix (Zanno et al., 2015; Drymala & Zanno, 2016) recover Kayentasuchus as the sister taxon of Crocodyliformes, with one of the synapomorphies that support this topology being the maxilla ventral to the antorbital fossa is of constant depth. However, the maxilla beneath the antorbital fenestra of Kayentasuchus is not of constant depth, it tapers posteriorly (Fig. 15B) (UCMP 131830; Clark & Sues, 2002). This character is described as: “Maxilla, posterior portion ventral to the antorbital fenestra: (0) tapers posteriorly; (1) has a similar dorsoventral depth as the anterior portion ventral to the antorbital fenestra; (2) expands dorsoventrally at the posterior margin of the maxilla” (character 27; pp. 65–66). This character was recently revised slightly by Butler et al. (2014) but the scoring in crocodylomorphs was not changed. However, this character has some issues regarding the definition of its states. In particular the difference between the states “taper posteriorly” and “approximately constant in dorsoventral depth” is generally not distinct (Fig. 15). Taxa such as Dromicosuchus (Sues et al., 2003), Hesperosuchus agilis (CM 29894) (Fig. 15D) and Junggarsuchus (Fig. 15C) (IVPP V 14010; coded by Drymala & Zanno, 2016) were coded as having the condition “aproximately constant in depth” when the morphology is not significantly different from the one present in Litargosuchus (BP/1/5237) or Dibothrosuchus (Fig. 15A) (IVPP V 7907), which were scored as tapering posteriorly. Additionally, other taxa like Sphenosuchus (Nesbitt, 2011) and Carnufex (Zanno et al., 2015; Drymala & Zanno, 2016) were scored as having maxillae with approximately constant depths. However, in both taxa this region is incompletely preserved (SAM-PK 3014; Drymala & Zanno, 2016), and this character cannot be scored. Finally, the antorbital fenestra of crocodyliforms is much smaller than that of non-crocodyliforms, and it is not clear that the depth of the maxilla ventral to it is comparable in taxa with fenestrae of different sizes. After evaluating the morphology of non-crocodyliform crocodylomorphs, the only taxon that has a markedly different maxillary morphology ventral to the antorbital fossa is Kayentasuchus (Fig. 15B). In Kayentasuchus (UCMP 131830) the anterior part of the maxilla ventral to the antorbital fossa is much taller dorsoventrally than the posterior part, as the anterior height is almost twice that of the posterior part. This morphology is unique to Kayentasuchus and seems to be an autapomorphy of the taxon. Thus, given the issues highlighted above we decided not to include this character in our analysis.

Figure 15 Detail of the region surrounding the antorbital fenestra with emphasis on the maxilla.

(A) Dibothrosuchus (IVPP V 7907); (B) Kayentasuchus (UCMP 131830); (C) Junggarsuchus (IVPP V 14010); and, Hesperosuchus (CM 29894). The border of the antorbital fossa is marked with a black dashed line, while the anterior and the posterior depths of the maxilla ventral to the antorbital fossa are marked with a white dashed line.

Phylogenetic analysis

The resulting dataset was analyzed using TNT v 1.5 (Goloboff, Farris & Nixon, 2008a; Goloboff, Farris & Nixon, 2008b). Unlike most previous analyses using this dataset, we used Gracilisuchus to root the trees given that it was recently recovered as one of the basal most suchians (Nesbitt, 2011; Butler et al., 2014). The analysis consisted of a heuristic search using 1000 RAS, saving up to ten equally parsimonious trees per replicate (hold = 10), and the trees found were subjected to a final round of TBR. Ten most parsimonious trees (MTPs) were recovered (L = 348; CI = 0.460; RI = 0.778), found in 735 of the 1,000 replicates. The branch swapping of these trees found no additional MPTs, securing that all the optimal trees were recovered in the analysis. The only difference among the MPTs is related to the relationships among the three notosuchians included in the dataset and the relationships among basal crocodylomorphs involving the “large bodied crocodylomorphs” (Fig. 16). However, the polytomy recovered in our analysis at the base of Crocodylomorpha is similar to the one obtained by Drymala & Zanno (2016), with the addition that Erpetosuchus is also involved in this uncertainty.

Figure 16 Time calibrated phylogenetic tree of the analysis performed in this contribution.

Strict consensus of the ten MTPs recovered (L = 348; CI = 0.460; RI = 0.778), found in 735 of the 1,000 replicates.

Unlike previous analyses using this dataset (Clark et al., 2004; or with modifications in Pol et al., 2013) we recovered a well-resolved topology for basal crocodylomorphs. Also, consistent with previous more inclusive datasets (Nesbitt, 2011; Drymala & Zanno, 2016) Saurosuchus was recovered at a more basal position than Postosuchus, and the latter (representing the rauisuchids) is the sister taxon of crocodylomorphs + Erpetosuchus + CM73372 + Redondavenator + Carnufex. Our results support the hypothesis of most recent papers with a focus on the relationships of Crocodylomorpha (Clark et al., 2004; Nesbitt, 2011; Pol et al., 2013), as we recovered a paraphyletic Sphenosuchia. The taxon that is the focus of this contribution, Macelognathus, was recovered forming a group with Almadasuchus and Hallopus that is sister to Crocodyliformes, corroborating Pol et al.’s (2013) initial hypothesis (Fig. 16). The relationships among non-crocodyliform crocodylomorphs are consistent with most previous analyses (Clark et al., 2004; Pol et al., 2013), but contrast strongly with Nesbitt’s (2011) and Drymala & Zanno’s (2016) results for crocodylomorphs, which did not include Macelognathus, Almadasuchus, or Hallopus and only the latter authors included Junggarsuchus. In those analyses, Litargosuchus and Kayentasuchus (and also Terrestrisuchus, either in a polytomy with or as the sister group of Dibothrosuchus) were found as successive sister groups of Crocodyliformes, while we recovered these taxa in much more basal phylogenetic positions within Crocodylomorpha. However, and as in the more inclusive analyses mentioned above (Nesbitt, 2011; Drymala & Zanno, 2016), we also recovered CM 73372, Carnufex and Redondavenator at the base of Crocodylomorpha. Synapomorphies supporting this topology are going to be treated on the following section.

CM 73372, Redondavenator, and Carnufex are recovered as the most basal crocodylomorphs (using the stem-based definition of (Nesbitt, 2011)), placed in a basal polytomy with Erpetosuchus (Fig. 16). This lack of resolution is mainly due to the fragmentary nature of these taxa and the very little anatomical overlap between the known specimens of CM 73372, Redondaventor, Carnufex and the Erpetosuchus specimens. The elongate preacetabular process of the ilium is not mapped as a synapomorphy grouping CM 73372 and crocodylomorphs, as the unknown condition in Erpetosuchus of this character makes this ambiguous. Both synapomorphies retrieved in our analysis joining these taxa with crocodylomorphs are absent in CM 73372, as these are cranial: maxillae meet to form a secondary palate (char. 3-1; Clark, 1994) and the absence of postfrontal bones (char. 8-1; Benton & Clark, 1988). A group of large pseudosuchians has been recovered previously as basal crocodylomorphs (“large-bodied crocodylomorphs” sensu Drymala & Zanno, 2016). In our strict consensus the smaller bodied Erpetosuchus is retrieved as a basal crocodylomorph, however this result should be considered preliminary, as our dataset lacks an extensive sampling of basal pseudosuchians and a detailed revision of the basal-most crocodylomorphs is still needed.

The derived features shared by taxa traditionally recognized as crocodylomorphs have been discussed extensively in various contributions (Walker, 1970; Clark, 1986; Benton & Clark, 1988; among many others latter papers), and these will not be discussed here. However, typical crocodylomorph anatomical features are not recovered as synapomorphies in our anaylsis (e.g., the postglenoid process of the coracoids, elongated proximal carpals (see Supplemental Information)) due to the missing data in Erpetosuchus and the “large bodied” crocodylomorphs causing ambiguous optimizations of those characters. A single synapomorphy supports the node that groups the classic or “small-bodied” taxa plus crocodyliforms, which is the lack of an ectepicondylar groove in the humerus (char. 136-1; Benton & Clark, 1988).

Unlike previous analyses using this dataset (Clark et al., 2004; Pol et al., 2013) greater resolution is observed in the strict consensus. However, this topology should be treated with caution, as many nodes are supported by a single synapomorphy, and the incorporation of new material or more characters might alter the topology. Near the base of Crocodylomorpha is a clade composed of Pseudhesperosuchus and Trialestes (Fig. 16). A basal position for Pseudhesperosuchus has been previously suggested (Benton & Clark, 1988; Wu & Chatterjee, 1993; Sues et al., 2003; Lecuona, Ezcurra & Irmis, 2016), but its grouping at the base with Trialestes is new to this analysis. This clade is diagnosed by a single synapomorphy: the lack of a well developed olecranon process (char. 117-1). The basal position of these taxa is given by the presence of elongated anterior parts of the facial portion of the maxillae, unlike the condition in more derived forms where this region is shorter than the posterior one (char. 2-1; Clark, 1994).

One node closer to crocodyliforms a clade composed of Saltoposuchus, Terrestrisuchus and Litargosuchus is recovered (Fig. 16). This clade could be identified as Saltoposuchidae Crush, 1984, if further corroborated. As previously mentioned this group is characterized by a single synapomorphy, which is a distally projected fibular condyle of the femur with respect to the medial one (char. 94-1; Pol et al., 2013). Terrestrisuchus and Litargosuchus are nested within this clade, more closely related than with Saltoposuchus, and this topology is supported by the following synapomorphies: a ridge along the border of the supratemporal fossa on the squamosal (char. 12-1; Clark, 1994); medial margins of the supratemporal fossae separated by a broad, flat area (char. 18-1; Clark, 1994); and, a very wide anterior process of the squamosal in dorsal view (char. 109-1).

A clade of North American crocodylomorphs is recovered as more closely related to crocodyliforms than the clade formed by Saltoposuchus, Terrestrisuchus and Litargosuchus (Fig. 16). The clade is formed by Hesperosuchus, Dromicosuchus and Kayentasuchus, with the inclusion of the latter being new to this analysis as it was recovered as the sister taxon of Crocodyliformes in some recent analyses (Nesbitt, 2011; Drymala & Zanno, 2016). The presence of a crest on the dorsal surface of the frontal (char. 9-1; Clark, 1994), a ventral exposure of the splenials on the mandibular symphysis (char. 84-1; Ortega, Buscalioni & Gasparini, 1996), and an elongated posterodorsal process of the postorbital (char. 101-1) support this group as closer to crocodyliforms. This clade of North American crocodylomorphs is diagnosed by the presence of a dorsomedial projection of the articular (char. 27-1; Clark, 1986; paralleled in Protosuchus and unknown in Kayentasuchus), and a well-defined antorbital fossa all around its circumference (char. 137-1; Drymala & Zanno, 2016). Dromicosuchus and Kayentasuchus are more closely related based on femoral characters: the presence of a pseudointernal trochanter (char. 92-1; Pol et al., 2013) and a ridge-like fourth trochanter (char. 93-1; Pol et al., 2013).

The topology that groups Sphenosuchus, Dibothrosuchus, Junggarsuchus and hallopodids with crocodyliforms is quite stable in this dataset and has been recovered in other analyses (Clark et al., 2004; Pol et al., 2013), even when the more basal nodes were very badly resolved. This contrasts with the more basal position recovered in Nesbitt’s analysis (and analyses derived from that study (e.g., Lecuona, Ezcurra & Irmis, 2016)), where Sphenosuchus was placeded more basally within Crocodylomorpha. The presence of completely sutured parietals (16-2; Clark, 1994), a sagittal crest separating the medial margin of the supratemporal fossae (char. 18-1; Clark, 1994), and a coracoid with a very elongated postglenoid process (char. 29-2; Clark, 1994) support this clade of derived crocodylomorphs. As in previous analyses (Pol et al., 2013), Junggarsuchus is recovered as the sister group of hallopoids (see below) + Crocodyliformes. This phylogenetic position is supported by the following synapomorphies: an expanded basisphenoid (char. 35-1; Clark, 1994); the exoccipitals contact the quadrate, enclosing the passage for the interal carotids (char. 36-1; Clark, 1994); large post-temporal fenestrae enclosed by the squamosal and the exoccipitals, with the lateral edge close to the supraoccipital (74-1; Pol et al., 2013); and, a radius shorter than the humerus (107-1). The later is new to this analysis and is paralleled in Saltoposuchus and variably present in Hesperosuchus (see Character Sampling).

Almadasuchus is found forming a clade with Macelognathus and Hallopus, the Hallopodidae, which is the sister group of Crocodyliformes (Fig. 16). Most of the characters that support the position of this clade as the sister group of Crocodyliformes were already reviewed by Pol et al. (2013), who did not include Macelognathus and Hallopus in their analysis, nonetheless we identified three new characters supporting this topology: a wide supraoccipital in posterior view (char. 98-1), a quadrate-laterosphenoid contact (char. 99-1), and the cranial nerves IX–XI exiting through a common foramen (foramen vagi, char. 105-1). This clade of Late Jurassic forms can be diagnosed by the following synapomorphies: the presence of a cranioquadrate passage that is not in the lateral border of the skull (char. 77-2; Clark, 1994); femoral head and distal condyles of the femur having parallel long axes (char. 90-1; Pol et al., 2013); a trochanteric crest on the femur (char. 91-0; Pol et al., 2013); and, a pseudointernal trochanter in the posterolateral end of the proximal end of the femur (char. 92-1; Pol et al., 2013; paralleled in Kayentasuchus + Dromicosuchus). Unlike the Argentinean form, Macelognathus and Hallopus share the reduction of the fourth trochanter to a sharp ridge (char. 93-1; Pol et al., 2013; paralleled in Kayentasuchus + Dromicosuchus).

Support measures are all low, exhibiting Bremer values of one and jackknife/bootstrap values well below 50 on the internal nodes of Crocodylomorpha. However when the different positions of very incomplete taxa like Carnufex, Redondavenator, and Trialestes are ignored in the calculations, support measures increase significantly. Also if the different phylogenetic postions of Litargosuchus are ignored in suboptimal trees support measures increase significantly, as it was noted above that with two extra steps this taxon was positioned as the sister group of Crocodyliformes. Under these circumstances the nodes that display a higher support (Bremer support = 3) are the one grouping the “small-bodied” crocodylomorphs, the one grouping the Jurassic non-crocodyliform crocodylomorphs (Sphenosuchus + Dibothrosuchus + Junggarsuchus + hallopoids + Crocodyliformes), Hallopodidae + Junggarsuchus + Crocodyliformes and other crocodyliform internal nodes. Another relatively well-supported node is the one grouping the North American crocodylomorphs (Hesperosuchus + Dromicosuchus + Kayentasuchus), as it shows Bremer values of 2. Unfortunately, the Hallopodidae only has a Bremer support of one (1), as in some suboptimal trees Almadasuchus is depicted as the sister taxon of Crocodyliformes

To test the robustness of our analysis we considered different topologies obtained in recent analyses. First, we tried testing a monophyletic Sphenosuchia (Bonaparte, 1972b), forcing all non-crocodyliform crocodylomorph taxa to form the sister clade of Crocodyliformes. In order to retrieve such a monophyletic group (i.e., a clade including all non-crocodyliform crocodylomorphs) nine to eighteen extra steps are required, depending on whether CM 73372 is part of this artificial clade. Thus, the idea of a monophyletic Sphenosuchia is highly suboptimal in this dataset. On the other hand, our analysis differs from the results recently obtained by Nesbitt (2011), Zanno et al. (2015) and Drymala & Zanno (2016). In these analyses Terrestrisuchus, Litargosuchus and Kayentasuchus were found as very closely related to Crocodyliformes, in particular the latter two where found as successive sister groups of Crocodyliformes. In our data set to set either Terrestrisuchus or Kayentasuchus as sister group to Crocodyliformes implies highly suboptimal hypothesis, as they require 10 and 8 extra steps respectively. However, a similar position solely for Litargosuchus is not a highly suboptimal topology, as 1 extra steps are required to place this taxon as the sister taxon of Crocodyliformes. This hypothesis finds support from the conditions present in the squamosal (broad anterodorsal process), parietal (broad area separating the supratemporal fossae) and postcranial skeleton (the postglenoid process), among other characters, of Litargosuchus. Unfortunately, due to the preservation of the only known specimen, no data can currently be retrieved on the braincase of the specimen.

Finally, Wilberg (2015) has recently underscored the importance of taxon sampling of basal crocodylomorphs for inferring the position of thalattosuchians, noting the latter can be retrieved as the sister group of Crocodyliformes (rather than within it) in some cases. A critical review of Wilberg’s data set is beyond the scope of the present paper but we have tested the impact of alternative positions of Thalattosuchia in the phylogeny of basal Crocodylomorpha within the context of our dataset. The basal position for Thalattosuchia as the sister group of Crocodyliformes requires 13 extra steps (L = 361). Forcing this topology also implies great changes in the strict consensus, as under this constraint we recover a large clade formed by almost every “sphenosuchian” with the exception of Litargosuchus, Pseudhesperosuchus, Erpetosuchus and the “large bodied crocodylomorphs” (see Supplemental Information). However, it is important to note that two clades still remain from the optimal topology: the North American clade (Hesperosuchus + Dromicosuchus + Kayentasuchus) and Hallopodidae (with the possible inclusion of Junggarsuchus to this clade). A further similarity to the unconstrained analysis is the close relationship between the Sphenosuchus, Dibothrosuchus and the hallopodids + Junggarsuchus, however these do not form a monophyletic clade with the crocodyliforms as in the optimal topology.

Discussion

Taxonomic status of Macelognathus vagans

The taxonomic status of Macelognathus vagans has been subject to debate among recent years. Most of these questions are based on two main topics: the similarities between Macelognathus vagans Marsh, 1884 and Hallopus victor Marsh, 1881; and, the provenance of the latter taxon. The similarities between Macelognathus and Hallopus were noted by Göhlich et al. (2005) when they described new postcranial elements assigned to Macelognathus. However most of the similarities noted (e.g., lack of supraacetabular crest) are widely present among non-crocodyliform crocodylomorphs. On the other hand, differences noted by these authors are based on proportions of certain elements that are very difficult to compare, such as the lateromedial development of the calcaneal tuber. The only trait that might shed some light on this matter (a dorsoventrally oriented crest on the medial surface of the calcaneal tuber) is not exposed in Hallopus, precluding any significant comparison.

The provenance of Hallopus has been a matter of dispute since it was originally published. As clearly shown by Schuchert (1939) and Walker (1970), neither Marsh nor his collector Baldwin retrieved the specimen themselves. Baldwin bought the specimen from a collector in Colorado Springs (Colorado, USA) and later checked the outcrops for additional material. Based on letters from Baldwin to Marsh, Shuchert (1939) concluded that the type specimen of Hallopus was retrieved from “The Nipple” area, which exposes a horizon assigned to the top of the Morrison Formation. However, subsequently Norell & Storrs (1989) challenged this view claiming that no beds such as the ones where Hallopus was found were known anywhere in the Morrison Formation. Thus, these authors suggested that it may come from levels of the underlying Ralston Formation (Callovian) from Oil Creek Canyon. Finally, Ague, Carpenter & Ostrom (1995) revisited “The Nipple” locality and recognized similar brownish/red levels in the top of the Morrison Formation. Furthermore, petrographic analyses showed that their samples and the ones from the matrix of the Hallopus specimen were almost identical (poorly sorted arkosic wackes). As a result they concluded that Hallopus was recovered from the upper levels of the Morrison Formation (Ague, Carpenter & Ostrom, 1995). Accepting these observations makes both taxa (Macelognathus and Hallopus) roughly coincident both in time and space (although the Fruita specimen is from lower in the upper part of the Morrison Formation and about 300 km away from The Nipple), facilitating the synonymy hypothesis.

Unfortunately, from our contribution little new information can be added to solve this problem as we describe the braincase of Macelognathus, a region unknown in Hallopus. Nevertheless, our results do find Macelognathus and Hallopus as sister taxa. Given the similarities and the occurrence of both taxa in the same formation, the potential synonymy (Macelognathus vagans Marsh, 1884 being a junior synonym of Hallopus, Marsh, 1881) has to be considered. Further preparation of the only known specimen of Hallopus (YPM 1914) or more material of this taxon, especially crania, would help to solve this uncertainty.

Phylogenetic position of Macelognathus and its implications

In spite of the low support in our analysis in some nodes of the consensus tree, some conclusions about the general pattern of the results can be extracted. Unlike some other modern crocodylomorph phylogenetic analyses (Clark et al., 2004; Pol et al., 2013) a well resolved phylogenetic hypothesis is here recovered, although with low support values. This is also valuable when it is considered that our dataset includes the largest number of non-crocodyliform crocodylomorphs than in any other analysis (e.g., Clark, Sues & Berman, 2000; Clark et al., 2004; Nesbitt, 2011; Pol et al., 2013; Zanno et al., 2015; Drymala & Zanno, 2016).

Among basal crocodylomorphs, especially near the base of the clade, a general pattern of small clades that soon go extinct can be recognized. Also these clades have a very important geographic influence, as basically Argentinean (Pseudhesperosuchus + Trialestes) and North American (Hesperosuchus + Dromicosuchus + Kayentasuchus) groups can be recognized. The former is positioned as the most basal clade among crocodylomorphs, while the latter is more closely related to derived crocodylomorphs such as Sphenosuchus and Dibothrosuchus. More basally a clade of mainly European forms is recognized (Terrestrisuchus and Saltoposuchus, which are sometimes synonymized), with the incorporation of the South African taxon Litargosuchus. All these clades represent several diversification pulses that took place in the Late Triassic among crocodylomorphs. While the basal clade formed by Trialestes and Pseudhesperosuchus is restricted to the Late Triassic, the other basal clades extend well into the Early Jurassic, in both cases represented by the most specialized members of the clade (Litargosuchus and Kayentasuchus).

Among the basal crocodylomorphs closest to Crocodyliformes we recovered a clade formed by the Late Jurassic Almadasuchus + Macelognathus + Hallopus, and its monophyly is not sensitive to the position of Thalattosuchia within Crocodylomorpha in the context of our dataset. Although their anatomy is very incompletely known, the Hallopodidae is characterized by a reduction of cranial kinesis (Pol et al., 2013) and several cursorial modifications (as highly elongated radiale, well projected femoral head, presence of additional trochanters on the femora). This clade of derived crocodylomorphs shares many derived traits also present in basal crocodyliforms (e.g., wide supraoccipitals, a quadrate laterosphenoid contact (see Phylogenetic analysis)) and some other traits present in more derived crocodyliforms (such as the presence of a cranioquadrate passage and the posterior closure of the otic notches). The phylogenetic position of this group as the sister group of Crocodyliformes implies a large amount of unsampled record (ghost lineage (Norell, 1992)) due to the presence of the crocodyliform Hemiprotosuchus in the Late Triassic (Kent et al., 2014). Thus, many traits recovered as synapomorphies of this group of Late Jurassic crocodylomorphs could be product of parallelisms with Crocodyliformes. More Jurassic crocodylomorphs remains, especially for the Middle and Late parts of this period, will be crucial to evaluate the evolution of these forms.

Conclusions

A partial braincase assigned to Macelognathus vagans Marsh, 1884 is described in this contribution. The use of computed tomography allowed us to study the internal morphology of this specimen and enabled us to identify structures that otherwise would not be visible. Furthermore, this study represents the first non-crocodyliform crocodylomorph that has been subjected to CT scanning. Scans of other crocodylomorph braincases will undoubtedly reveal similar complexity and should be attempted soon.

Although incomplete, the braincase of Macelognathus has many traits that allowed us to recognize its affinities among derived crocodylomorphs (e.g., an expanded and pneumatized basisphenoid; a pneumatized quadrate; a quadrate-otoccipital contact; a wide supraoccipital; a cranioquadrate passage). In particular, Macelognathus was recovered as a member of a highly derived group of Late Jurassic crocodylomorphs together with Almadasuchus from Argentina and Hallopus from the USA. These forms are highly cursorial, have large otic notches and a higher integration of the quadrate with their braincases when compared with more basal forms. Their position as the sister group of crocodyliforms is very interesting, as they allow a better understanding of the transformations from the typical kinetic skulls of reptiles (Holliday & Witmer, 2008) to the typical akinetic skulls of extant crocodylians and crocodyliforms (Langston, 1973; Pol et al., 2013).

Finally, we compiled a data set incorporating most non-crocodyliform crocodylomorphs known to date. Many historically important taxa were incorporated, whose phylogenetic position was not previously tested before (Trialestes, Hallopus, and Macelognathus). Although some groups have low support, we recovered a well-resolved phylogenetic hypothesis when compared to previous studies. In our analyses we found several small basal clades, following a recognizable geographic pattern, and a relatively stable group of crocodylomorphs closely related to crocodyliforms. Macelognathus was found grouped with two other Late Jurassic crocodylomorphs (Almadasuchus and Hallopus), and this group was found as the sister group of crocodyliforms.

Supplemental Information

Supplemental Information 1 Supplementary Information

Includes specimen list, character list, additional cladograms depicting the support measures and the synapomorphy list for each node.

Click here for additional data file.

Supplemental Information 2 TNT file of the data matrix used in this contribution

Click here for additional data file.

We would like to thank S Goldberg, M Hill and H Towbin from the Microscopy and Imaging Facility of the American Museum of Natural History for their valuable help during the CT process and M Norell for arranging for us to use it. The Academic Editor (M Young) and two anonymous reviewers are thanked, as their suggestions greatly increased the final quality of the resulting manuscript. We are also thankful to LM Chiappe for allowing us to study this specimen under his care, and M Walsh for helping with the loans of the specimen of Macelognathus. We are grateful to P Mocho (LACM) for his help with the photos of the type specimen. Finally, K Poole (GWU) and MA Ordoñez (IDEAN) are thanked for their help in the posterior processing of the 3D meshes. TNT is a free program made available by the Willi Hennig Society. This is JML’s R-198 contribution to the Instituto de Estudios Andinos Don Pablo Groeber.

Abbreviations

AMNH American Museum of Natural History (Fossil Reptiles), New York, United States

NHMUK Natural History Museum, London, UK

BSP Bayerische Staatssammlung fur Palaontologie und Geologie, Munich, Germany

BP Evolutionary Studies Institute (formerly Bernard Price Institute for Palaeontological Research), University of the Witwatersrand, Johannesburg, South Africa

CM Carnegie Museum of Natural History, Pittsburg, United States

IVPP Institute of Vertebrate Paleontology and Paleoanthropology, Chinese Academy of Sciences, Beijing, China

LACM Natural History Museum of Los Angeles County, Los Angeles, United States

MACN Museo Argentino de Ciencias Naturales, Buenos Aires, Argentina

MCZ Museum of Comparative Zoology, Cambridge, United States

MLP Museo de La Plata, La Plata, Argentina

MPEF Museo Paleontológico Egidio Feruglio, Trelew, Argentina

PVL Museo Miguel Lillo, San Miguel de Tucumán, Argentina

PVSJ Museo de Ciencias Naturales, Universidad Nacional de San Juan, San Juan, Argentina

SAM Iziko South African Museum, Cape Town, South Africa

SMNS Staatliches Museum fur Naturkunde, Stuttgart, Germany

UCMP University of California Museum of Paleontology, Berkeley, United States

UFGRS Universidade Federal do Rio Grande Do Sul, Porto Alegre, Brasil

UFRJ Universidade Federal do Rio do Janeiro, Rio do Janeiro, Brasil

NCSM North Carolina Museum of Natural Sciences

YPM Yale Peabody Museum, New Haven, United States.

Additional Information and Declarations

Competing Interests

Author Contributions

Data Availability

The authors declare there are no competing interests.

Juan Martin Leardi conceived and designed the experiments, performed the experiments, analyzed the data, wrote the paper, prepared figures and/or tables, reviewed drafts of the paper.

Diego Pol conceived and designed the experiments, wrote the paper, prepared figures and/or tables, reviewed drafts of the paper.

James Matthew Clark wrote the paper, prepared figures and/or tables, reviewed drafts of the paper.

The following information was supplied regarding data availability:

Morphobank: http://dx.doi.org/10.7934/P2550.

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
