# Peer review of "Detailed anatomy of the braincase of Macelognathus vagans Marsh, 1884 (Archosauria, Crocodylomorpha) using high resolution tomography and new insights on basal crocodylomorph phylogeny"

_PeerJ, doi:10.7717/peerj.2801_

## Round 0.1 · original submission · Minor Revisions

Dear authors,

I have accepted the decision of ‘minor revisions’ from the two reviewers.

Furthermore, I have some additional comments that the authors should address prior to resubmission (in addition to those made by the reviewers):

1. Authority and date should be provided for each species-level taxon at first mention (including in the title, abstract and introduction). Please ensure that the nominal authority is also included in the reference list.
2. In your revised species diagnosis, can you indicate which characteristics are autapomorphic.
3. Can you add in a diagnosis for Hallopoidae.
4. Please replace 'trees' with 'cladograms' or ‘topology’ where appropriate.

Once again, thank you for submitting your manuscript to PeerJ and I look forward to receiving your revision.

Reviewer 1 ·

Basic reporting

The submitted article describes new braincase material of the crocodylomorph Macelognathus. The writing is direct, well-constructed and organized with only a few issues. Figure 1 is out of focus and the paper requires a figure (Fig 2) showing the raw CT data.

All of the sundry comments are below.
Ln 77: missing “History”…AMN H.
Ln 297: the passage on the ‘otic joint’. I think here you are describing the orbital process of the quadrate, not the otic joint, or what you call the ‘primary head of the quadrate’. The otic joint is specifically the articulation between the otic process of the quadrate and the squamosal/exoccipital region and is homologous with the otic joints of other sauropsids. This is the part that doesn’t really ever suture with the skull roof in crocs (or birds and lizards). On the other hand, the orbital process of the quadrate is the portion that contacts and variably sutures to the prootic. It lies between the pterygoid ramus and the otic process of the quadrate.

Ln 363 (and character 104): rather than saying the prootic is verticalized on the supratemporal “fenestra”, isn’t this really the supratemporal fossa? Whereas the fenestra is simply the 2-dimensional window into the skull, the ‘fossa’ is the surface within it where the muscles and other soft tissues attach. I understand how the character is based on ‘viewing’ the bone through the fenestra, and that its visible…but the prootic is not really contributing any morphology to the actual fenestra…just the fossa. I suppose this is how this character has often been described in the literature, but it could be phrased more correctly. For example, the character should properly read: “Prootic: exposed in dorsal view, through the supratemporal fenestra”…not “on” the fenestra.
Ln 588: Crocodylomorphs..spelling
Ln 662: Carnufex repeated..

Ln 1130: similar argument as the other fenestra/fossa description. As I understand it, the antorbital fenestra is the 2 dimensional window outlined by the outer surfaces of the facial bones whereas the fossa (or cavity) lies deep, or just medial to the fenestra. So I don’t understand 137 (1) where the Fossa can form a complete circumference “around” the Fenestra…it seems at odds with the terminology. Perhaps the fossa is “within” the Fenestra? This would better correspond with Figure 15.
Ln 664: Sentence is awkward, please rephrase

Fig 1. I know the fossil is a scrappy thing, but the photographs are not in good focus and only show two views. The second has the quadrate out of focus. There are no oblique, caudal, ventral or other complementary views. The illustrations are nice, but more photos are appreciated.
Fig ?. It is important to show examples of the CT slice data. These are the data that your 3D models/hypotheses are derived from and it is critical to show the readers what kind of contrast levels you were making your morphological hypotheses with. It is also important to show and label the important elements in the CT data.

Experimental design

No Comments

Validity of the findings

No Comments

Additional comments

No Comments

Reviewer 2 ·

Basic reporting

No Comments

Experimental design

No comments

Validity of the findings

No comments

Additional comments

Overview:
Leardi and colleagues describe in great detail the braincase of Macelognathus vagans from micro-CT data – a taxon known previously almost exclusively from mandibular and postcranial material. Then, using this information, they present a new phylogenetic analysis including the largest sample yet of basal crocodylomorphs and newly developed characters. They find Macelognathus as part of a clade of North and South American Late Jurassic taxa forming the proximal outgroup to Crocodyliformes. They then briefly discuss the implications of this new phylogeny for character evolution and biogeography in Crocodylomorpha. I feel this submission is already well polished and will require only minor revisions to be publishable in PeerJ. I also appreciate the level of anatomical detail described in this manuscript and the abundant comparisons with other crocodylomorph material. Many descriptions of “sphenosuchian” crocodylomorphs have been frustratingly brief and inadequately figured. This submission is something of a breath of fresh air in the vein of Walker’s Sphenosuchus monograph (if a long, detailed description of the minutia of braincase anatomy can be considered a ‘breath of fresh air’).

General comments: (line numbers are from the review pdf)

Line 68 – “computer” should be “computed”

Lines 81-82 – “slice archive data is available upon request…” – It would be nice if this data could be made available through an online repository such as Dryad, or MorphoSource.

Lines 90-91 – The authors define “Hallopodidae” as the descendents of the closest common ancestor of Hallopus, Macelognathus, and Almadasuchus. While this works for the presented phylogeny, defining the family in this way means it could easily become much more inclusive if one of the included taxa shifts to even a slightly different part of the tree in future analyses. Maybe it would be more prudent to define it as something like “all taxa more closely related to Hallopus” than to some other chosen taxon?

Line 114 – “cranioquadrate passage present” – the words “fully enclosed” could be added here. While one could argue that “passage” suggests an enclosed structure, terms like “open cranioquadrate passage/canal” have been used in the literature (e.g. Turner, 2015).

Line 115 – “development” could be replaced by “height”

Line 115 – “quadrate contacts the supraoccipital” – Maybe the authors meant “otoccipital” here? According to both the description and figures in this manuscript, the quadrate does not contact the supraoccipital.

Line 116 – “posterior bony ring on the basisphenoid” – maybe add a little more detail to this feature to better indicate which part of the basisphenoid they are referring to – maybe something like “allowing the precarotid recess open posteriorly”?

Lines 154-156 – “the prefrontal contacts medially with the… frontal only, a rare condition among crocodylomorphs” – maybe change the wording here to clarify that it is possible that the nasals and prefrontals contacted each other, but only part of the prefrontal is preserved (and none of the nasals). This is mentioned in the next sentence, but this first sounds quite confident in the statement that the prefrontals do not contact the nasals.

Line 193 – “anteromedially” – should this be “anterolaterally”? – This is how it looks in figure 2.

Line 195 – “interpreted as the anterior border of the STF” – could this crest be labeled on Figure 3?

Line 207 – “… is preserved on both sides” - both sides? A fragmentary postoribtal is shown only on the right side in figures 2 and 3.

Line 239 – “Medial to this crest…” – Do the authors mean “posterior to”?

Lines 255-256 – “the otic aperture is not such a circumscribed fenestra” – I’m not sure what the authors mean by this. Perhaps elaborate here to better explain how the otic aperture differs in other taxa besides the size.

Line 335 – “This 4th chamber is also externally connected via the pneumatic quadrate foramen…” - in figure 5, the “quadrate fenestra” label points to space 3. Also, this feature is described as a “quadrate fenestra” in other regions of the manuscript and figure captions.

Line 409 – “crista protica” – should be “crista prootica”. This same misspelling is present in the figure captions.

Line 463-463 – “… medially projected process that excludes the basioccipital from the ventromedial border of the foramen magnum” – Do the authors mean “ventrolateral”? The basioccipital certainly contributes to the ventromedial border of the FM.

Lines 487-488 – “… unlike the condition seen in crocodyliforms” – Thalattosuchians also lack a transverse canal passing through the supraoccipital.

Line 497 – “one” should be replaced by “half”

Line 501 – missing open parenthesis before “Fig. 10B-C”
Line 547 – “posteriorly projected process of the basioccipital…” - is this reversed? Above it says a posterior process of the basisphenoid projects into the anterior surface of the basioccipital - "Anteriorly, the median region of the basioccipital is divided by a short posterodorsal process of the basisphenoid".

Line 641 – “new dataset…” – Is this dataset going to be reposited online (e.g. MorphoBank) or included as supplementary info for this paper? If not, it should be according to PeerJ policy.

Line 662 – the word “Carnufex” is repeated twice

Line 696 – a word is missing from this sentence. Either the word “the” needs to be inserted between “in” and “other”, or “crocodylomorph” needs to be “crocodylomorphs”.

Lines 716-721 - if the supraoccipitals are fused to the adjacent bones (as in the sutures are obliterated), how can one tell what the shape of the bone was? The authors could be a little more explicit as to how they are evaluating shape without visible sutures. Or, alternatively, if they mean something else by “fused”, they should explain that.

Lines 725-727 – “… crocodyliformes have a wide supraoccipital bone, where this element is wider than tall.” - some metriorhynchids have rather tall supraoccipitals with a height exceeding the width.

Line 744 – “… forms the posterior margin of the median Eustachian tube” – Is there any indication of a median eustachian tube in Macelognathus?

Line 1004 – “… nor in thalattosuchians” – This is not true of all thalattosuchians. In at least Steneosaurus bollensis and Platysuchus, the anterior process exceeds the acetabulum, looking very similar to the condition in Orthosuchus.

Line 1177 – “Carnufex” repeated twice

Line 1222 – “Crocodylomorphs” – should either be “Crocodylomorpha”, or should not be capitalized.

Line 1291 – “bellow” should be “below”

Lines 1352-1353 – “… Thalattosuchia as sister group of Crocodyliformes is highly suboptimal, requiring 48 extra steps…”. – I’m not sure if this result was left over from a different matrix iteration, but, based on the .tnt file provided for review, constraining Thalattosuchia to fall as sister to Crocodyliformes only adds 13 additional steps (L = 361). Perhaps also noteworty is that the only thalattosuchians sampled here are among the most derived – restricted to the bizarre metriorhynchids.

Line 1437 – Kent et al., 2014 – is missing from the reference list

Line 1447 – “ct” should be capitalized

References – I did not go over the references in any detail, but here are a few things I noticed:
– a large number of dashes in page number ranges need to be changed to em dashes
– Iordansky citation needs page numbers
– Leardi 2013 – “Leardi, 2012” is referenced a number of times in the text – I think 2013 is correct, so the in line citations should be corrected.
Figures:
Figure 1 – Parts A and C both have bad depth of field. It would be nice if this could be corrected. However, I don’t think this is too big of a deal as the drawings and CT models make up for it.

Figure 6 – A color key would be helpful on this figure so the reader wouldn’t have to refer back to figure 2. Also, the abbreviations g. csc (groove for semicircular canal?), aur (auricular recess) are not defined in the figure caption.

Figure 8 – “pcb” missing from figure caption. Also should "pcb" be "pcv" (posterior cerebral vein)?

---

## Round 0.2 · Minor Revisions

Dear authors,

The reviewer has decided upon 'minor revisions', which I have accepted. The reviewer also has a a .TNT file which has the constraints used in their review. I will email that directly to the lead author once I have a copy.

Reviewer 2 ·

Basic reporting

No Comments

Experimental design

No Comments

Validity of the findings

No Comments

Additional comments

Overall I am happy with all the revisions made by the authors and think this will make an excellent paper. I still have a couple of minor (and one major) comments. I think discrepancies with the tree lengths for the constrained topologies need to be addressed prior to publication. Other than that, I think it is basically ready to go.

Comments on response letter:

On the presence or absence of a transverse canal passing through the supraoccipital in thalattosuchians, I disagree with the authors that this condition is not adequately known. While it is true that it is not observable in the vast majority specimens, the sectioned Cambridge Teleosaurus braincase figured in Wilberg (2015) does demonstrate its absence (though I’m not necessarily claiming this is obvious in the figure). As do all CT scans of thalattosuchians of which I am aware (e.g. Metriorhynchus - Fernandez et al. 2011; Steneosaurus - Brusatte et al. 2016; Pelagosaurus - Dufeau 2011 PhD diss.; Pierce et al. PeerJ preprint). While it is not crucial for this point to be added to the manuscript, I do not think the statement that thalattosuchians lack this pneumatic feature is as controversial as the authors make it seem.

On the number of additional steps required to remove Thalattosuchia from Crocodyliformes (and other suboptimal topologies) – I did not simply cut and paste the thalattosuchian clade on a tree and look at the difference in tree length, but instead constrained a monophyletic Crocodyliformes excluding Thalattosuchia, and searched for MPTs while enforcing this constraint. As I typically use the GUI version of TNT, I followed the authors’ advice and ran the constraints in command line, using the ‘force’ command to ensure I wasn’t doing something incorrectly. This resulted in the same trees and tree length I recovered before – 361. Additionally, in looking closer at the authors’ supplementary info document, in the figure caption for figure 8S – the figure showing the topology with Thalattosuchia as sister to Crocodyliformes – the tree length is listed as 361, not 396 (and the topology is identical to what I recovered). I suggest that the authors investigate their code to determine where this discrepancy is arising as, based on this figure caption, they must have recovered this length on their own at some point. I checked the other reported step differences between MPTs and constrained topologies and found other minor disagreements. When constraining Terrestrisuchus to be the sister taxon to Crocodyliformes, I recover a tree length of 358 – 10 additional steps as opposed to the 12 steps reported. When constraining Litargosuchus to this position, I recovered a tree length of only 1 additional step (as opposed to the 2 reported). If possible, I will upload the .tnt files I modified implementing these 3 constraints with this review for the authors to check against their own.

On the use of dashes vs. em dashes in citations – there is still a mixture of dashes and em dashes used in page ranges. As the authors said, usage should be uniform.

Comments on manuscript:
Line 1383 – should “hallopoids” be “hallopodids”?

---

## Round 0.3 · accepted · Accept

Dear authors,

Many thanks for your revised manuscript. After reading it, I have accepted it for publication in PeerJ.

Once again, thank you for submitting your manuscript to PeerJ and I hope you will use us again as your publication venue.

If we need to clarify any details required to move the manuscript forward, then our production staff will get in touch with you. Otherwise, a proof will be forthcoming shortly for your review.

Congratulations and thank you for your submission.